# P2Y1 purinergic receptor identified as a diabetes target in a small-molecule screen to reverse circadian β-cell failure

**Biliana Marcheva[1†], Benjamin J Weidemann[1†], Akihiko Taguchi[1,2†], Mark Perelis[1,3], Kathryn Moynihan Ramsey[1], Marsha V Newman[1], Yumiko Kobayashi[1], Chiaki Omura[1], Jocelyn E Manning Fox[4], Haopeng Lin[4], Patrick E Macdonald[4], Joseph Bass[1]***

[1]Department of Medicine, Division of Endocrinology, Metabolism and Molecular Medicine, Northwestern University Feinberg School of Medicine, Chicago, United States; [2]Division of Endocrinology, Metabolism, Hematological Science and Therapeutics, Department of Bio-Signal Analysis, Yamaguchi University, Graduate School of Medicine, 1-1-1, Yamaguchi, Japan; [3]Ionis Pharmaceuticals, Inc, Carlsbad, United States; [4]Department of Pharmacology, Alberta Diabetes Institute, University of Alberta, Edmonton, AB, Canada

**\*For correspondence:**
j-bass@northwestern.edu

[†]These authors contributed equally to this work

**Reviewing Editor:**
Achim Kramer, Charite Universitaetsmedizin Berlin, Germany

**Abstract** The mammalian circadian clock drives daily oscillations in physiology and behavior through an autoregulatory transcription feedback loop present in central and peripheral cells. Ablation of the core clock within the endocrine pancreas of adult animals impairs the transcription and splicing of genes involved in hormone exocytosis and causes hypoinsulinemic diabetes. Here, we developed a genetically sensitized small-molecule screen to identify druggable proteins and mechanistic pathways involved in circadian β-cell failure. Our approach was to generate β-cells expressing a nanoluciferase reporter within the proinsulin polypeptide to screen 2640 pharmacologically active compounds and identify insulinotropic molecules that bypass the secretory defect in CRISPR-Cas9-targeted clock mutant β-cells. We validated hit compounds in primary mouse islets and identified known modulators of ligand-gated ion channels and G-protein-coupled receptors, including the antihelmintic ivermectin. Single-cell electrophysiology in circadian mutant mouse and human cadaveric islets revealed ivermectin as a glucose-dependent secretagogue. Genetic, genomic, and pharmacological analyses established the P2Y1 receptor as a clock-controlled mediator of the insulinotropic activity of ivermectin. These findings identify the P2Y1 purinergic receptor as a diabetes target based upon a genetically sensitized phenotypic screen.

## Editor's evaluation

Circadian disruption is widespread in our modern 24/7 society, leading to an increased prevalence of common diseases including type 2 diabetes. The authors conducted an unbiased screen for small-molecule compounds that can restore the attenuated insulin secretion from pancreatic β-cells caused by a disrupted circadian clock. They identified ivermectin and its clock-controlled target, the P2Y1 receptor, which regulate glucose-stimulated ca²⁺ influx and insulin secretion in β-cells. This discovery represents an important advance in our understanding of regulatory mechanisms of insulin secretion by cell-autonomous clocks in mouse and human β-cells and is of fundamental clinical importance in context of novel therapeutic targets for diabetes management.

## Introduction

Type 2 diabetes is an escalating epidemic involving gene-environment interactions that culminate in β-cell failure and insulin resistance. Recent epidemiological evidence has shown that shift work and sleep disturbance are environmental risk factors for diabetes (*Perelis et al., 2016*), while experimental genetic studies have shown that clock gene disruption within the endocrine pancreas causes hypoinsulinemic diabetes (*Marcheva et al., 2010*; *Sadacca et al., 2011*). At the molecular level, the circadian clock is composed of an autoregulatory transcriptional loop in which CLOCK/BMAL1 activate the repressors PER1/2/3 and CRY1/2, which feedback to inhibit CLOCK/BMAL1 in a cycle that repeats itself every 24 hr. An additional stabilizing loop involving ROR/REV-ERB regulates BMAL1 expression (*Kim and Lazar, 2020*). Recent chemical screens have identified new factors that modulate the core clock, including casein kinase 1 inhibitors that lengthen the circadian period through stabilizing PER proteins (*Hirota et al., 2010*; *Chen et al., 2012*), and a separate series of cryptochrome stabilizer compounds have been discovered that control glucose homeostasis in vivo (*Hirota et al., 2012*). Modulators of clock transcription factors may also control whole animal metabolism (*He et al., 2016*), though such compounds lack specificity (*Dierickx et al., 2019*).

Here, we developed a high-throughput small-molecule screen to identify insulinotropic compounds that act downstream of the circadian clock rather than through modulation of the core clock itself. We reasoned that compounds that enhance insulin secretion in the setting of β-cell clock disruption might in turn uncover therapeutic targets for more common forms of diabetes mellitus (*Marcheva et al., 2010*; *Perelis et al., 2015*; *Marcheva et al., 2020*; *Moffat et al., 2017*). To do so, we generated β-cells harboring a circadian gene mutation by CRISPR-Cas9 and co-expressing a luminescent insulin reporter that has previously been used to identify factors that either activated or repressed glucose-stimulated insulin secretion (GSIS) in wild-type β-cell lines (*Burns et al., 2015*). In our screen of 2640 drug or drug-like compounds in circadian mutant β-cells, we identified the macrolide ivermectin (IVM) as an insulinotropic compound that activates the P2Y1 purinergic receptor. We further identified the P2Y1 receptor as a direct transcriptional target of the molecular clock factor BMAL1 and a potent regulator of glucose-dependent calcium signaling. Our findings establish a chemical genetic strategy to identify novel endocrine cell therapeutics.

## Results

### High-throughput screen for chemical modulators of insulin secretion in circadian mutant β-cell

Based upon our finding that circadian genes regulate β-cell function, we developed a chemical genetic screen to identify pathways that enhance glucose-coupled insulin secretion in a cell-based model of circadian β-cell failure (*Figure 1A*). We previously showed that clonal *Bmal1*[-/-] Beta-TC-6 β-cell lines recapitulate the secretory defects observed in primary clock-deficient islets (*Marcheva et al., 2010*; *Perelis et al., 2015*; *Marcheva et al., 2020*). We next generated stable WT and *Bmal1*[-/-] β-cell lines with a luciferase readout for insulin secretion using an insulin-NanoLuciferase (NanoLuc)-expressing lentivirus (*Figure 1B*). We validated the direct correspondence between insulin-NanoLuc bioluminescence and levels of peptide secretion under increasing physiological concentrations of glucose (2–20 mM; $R^2 = 0.8937$; *Figure 1C*). We further confirmed impaired insulin secretion by reduced bioluminescence in *Bmal1*[-/-] compared to WT β-cell lines expressing insulin-NanoLuc in response to stimulatory concentrations of glucose (20 mM), potassium chloride, forskolin, and the phosphodiesterase inhibitor 3-isobutyl-1-methylxanthine (IBMX) (*Figure 1D*). We also validated the use of the DAG mimetic phorbol 12-myristate 13-acetate (PMA) as a positive control for the screen (*Figure 1D–F*; *Perelis et al., 2015*). A feasibility test with a Z'-factor score of 0.69 indicated a significant separation between the distribution of bioluminescent signal from the positive (10 µM PMA + 20 mM glucose) and negative (20 mM glucose) controls, suggesting that the assay provides a suitable platform for a high-throughput screen (*Figure 1F*; *Zhang et al., 1999*).

**eLife digest** Circadian rhythms – 'inbuilt' 24-hour cycles – control many aspects of behaviour and physiology. In mammals, they operate in nearly all tissues, including those involved in glucose metabolism. Recent studies have shown that mice with faulty genes involved in circadian rhythms, the core clock genes, can develop diabetes.

Diabetes arises when the body struggles to regulate blood sugar levels. In healthy individuals, the hormone insulin produced by beta cells in the pancreas regulates the amount of sugar in the blood. But when beta cells are faulty and do not generate sufficient insulin levels, or when insulin lacks the ability to stimulate cells to take up glucose, diabetes can develop.

Marcheva, Weidemann, Taguchi et al. wanted to find out if diabetes caused by impaired clock genes could be treated by targeting pathways regulating the secretion of insulin. To do so, they tested over 2,500 potential drugs on genetically modified beta cells with faulty core clock genes. They further screened the drugs on mice with the same defect in their beta cells.

Marcheva et al. identified one potential compound, the anti-parasite drug ivermectin, which was able to restore the secretion of insulin. When ivermectin was applied to both healthy mice and mice with faulty beta cells, the drug improved the control over glucose levels by activating a specific protein receptor that senses molecules important for storing and utilizing energy.

The findings reveal new drug targets for treating forms of diabetes associated with deregulation of the pancreatic circadian clock. The drug screening strategy used in the study may also be applied to reveal mechanisms underlying other conditions associated with disrupted circadian clocks, including sleep loss and jetlag.

## Identification and validation of high-throughput screen lead compounds in murine islets at high and low glucose concentrations

We next used insulin-NanoLuc-expressing $Bmal1^{-/-}$ β-cell lines to screen 2640 drugs and drug-like molecules from the Spectrum Collection (MicroSource Discovery Systems, Inc, New Milford, CT) to identify compounds that enhance insulin secretion (*Figure 1E*). Insulin-NanoLuc-expressing $Bmal1^{-/-}$ Beta-TC-6 cells were plated at 40,000 cells/well in a total of nine 384-well plates, incubated for 3 days, and then treated for 1 hr with either (i) 20 mM glucose alone (negative control that elicits reduced insulin secretion in $Bmal1^{-/-}$ cells), (ii) 20 mM glucose plus 10 μM of one of the 2640 compounds, or (iii) 20 mM glucose plus 10 μM PMA (positive control known to enhance insulin secretion in both $Bmal1^{-/-}$ mouse islets and Beta-TC-6 cells) (*Perelis et al., 2015*). Luciferase intensity from the supernatant was measured following exposure to NanoGlo Luciferase Assay Substrate (*Figure 1E*).

We initially identified 19 hit compounds that both significantly enhanced insulin secretion and elicited a response of greater than 3 standard deviations from the mean (Z-score > 3) with more than a 1.25-fold increase, exceeding the upper 99% confidence interval of the negative control (*Figure 2A*, *Figure 2—figure supplement 1A*, *Supplementary file 1*). Of these, seven were excluded from further analysis because of reported toxic effects or lack of availability of the compound (*Figure 2—figure supplement 1A*). The remaining 12 hit compounds mediate activity of ligand-gated cell surface receptors and ion channels that stimulate second messenger signaling cascades (*Figure 2B and C*; *Gaulton et al., 2010*; *Carrano et al., 2017*). Of these, four target ion channels (tacrine hydrochloride, suloctidil, dyclonine hydrochloride, and IVM) (*Figure 2B and C*; *Karlsson and Ahrén, 1992*; *Chatelain et al., 1984*; *Khanna et al., 2011*; *Chen and Kubo, 2018*; *Freeman et al., 1988*; *de Gaetano et al., 1976*; *Kornhuber et al., 2008*; *Sahdeo et al., 2014*; *Roghani et al., 1999*; *Ikeda, 2003*). Five target seven-transmembrane G-protein coupled receptors (GPCRs) that signal through phospholipase C (PLC) and diacylglycerol (DAG) to activate insulin secretion and β-cell gene transcription (benzalkonium chloride, carbachol, isoetharine mesylate, pipamperone, and IVM) (*Figure 2B and C*; *Chen and Kubo, 2018*; *Higashijima et al., 1990*; *Rinne et al., 2015*; *Bierman, 1983*; *Van Craenenbroeck et al., 2006*; *Nagata et al., 2019*; *Ratajewski et al., 2015*; *Ohtani et al., 2011*). Similar to the hit compounds of our screen, our previous results showed that carbachol, a muscarinic $G_q$-coupled receptor agonist, and the DAG mimetic PMA rescue insulin secretion in $Bmal1^{-/-}$ islets (*Perelis et al., 2015*). Four additional hit compounds act as acetylcholinesterase inhibitors, promoting enhanced glucose-dependent insulin

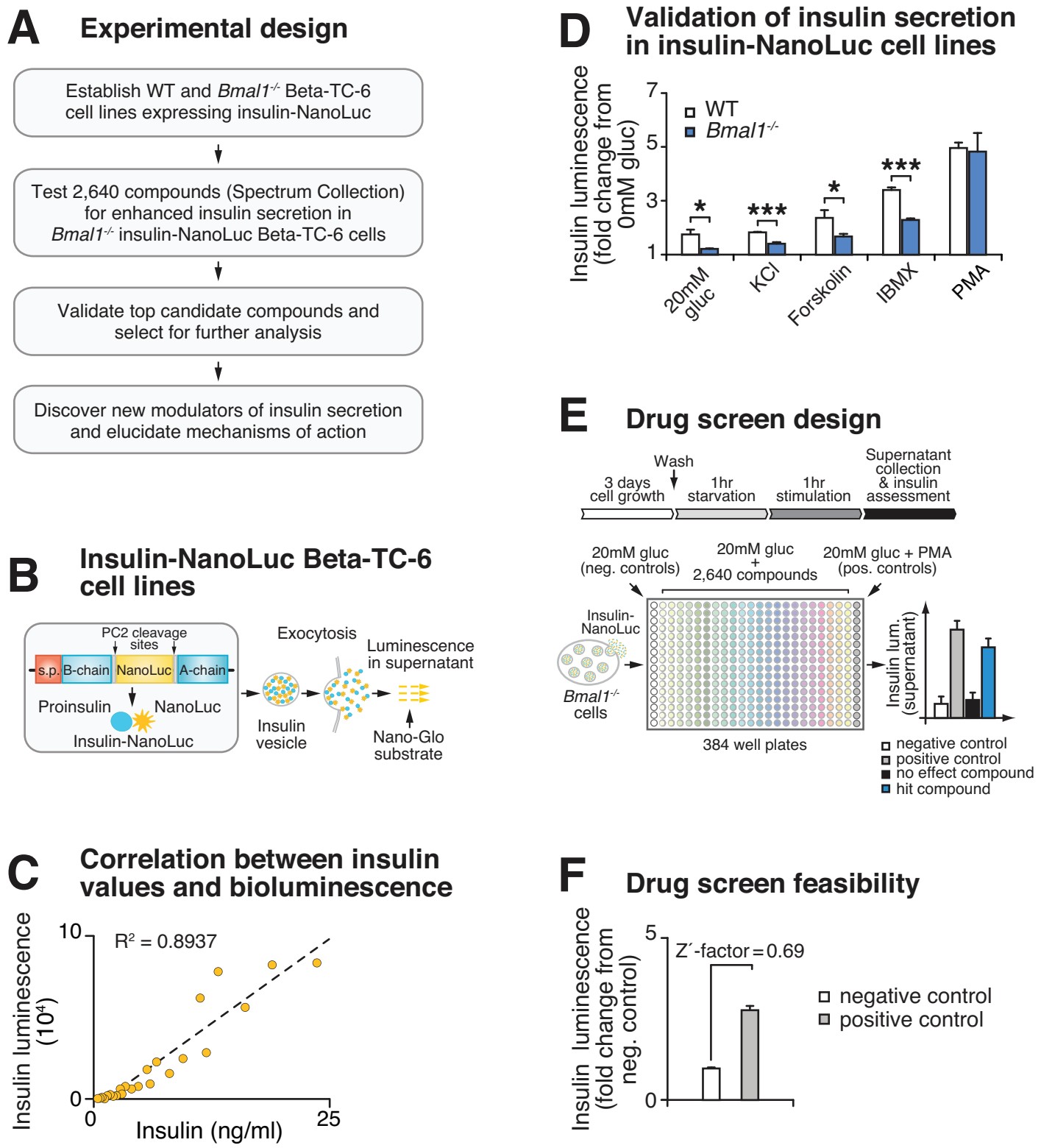

**Figure 1.** High-throughput screen for chemical modulators of insulin secretion in circadian mutant β-cells. (**A**) Flow chart of 'phenotype'-driven cell-based genetic screening platform to identify molecules and pathways that enhance insulin secretion during circadian β-cell failure. (**B**) Schematic of insulin-NanoLuciferase (NanoLuc) fusion construct, with bioluminescence detected in the supernatant as a proxy for insulin secretion. (**C**) Correlation between insulin-NanoLuc bioluminescence and insulin values measured by ELISA in response to a range of glucose concentrations (2–20 mM; $R^2$ =

*Figure 1 continued on next page*

*Figure 1 continued*

0.8937). (**D**) Insulin-NanoLuc bioluminescence following 1 hr exposure to 20 mM glucose, 30 mM KCl, and 20 mM glucose plus 2.5 µM forskolin, 500 µM 3-isobutyl-1-methylxanthine (IBMX), or 10 µM phorbol 12-myristate 13-acetate (PMA) in WT and *Bmal1*[-/-] insulin-NanoLuc Beta-TC-6 cells (n = 3–10 experimental repeats/condition). (**E**) Drug screen design. Insulin-NanoLuc-expressing Beta-TC-6 *Bmal1*[-/-] cells were plated in nine 384-well plates prior to exposure to 10 µM of each of the 2640 compounds from the Spectrum Collection in combination with 20 mM glucose. Negative (20 mM glucose alone) and positive (20 mM glucose plus 10 µM PMA) controls were included on each plate. (**F**) Drug screen feasibility test comparing negative (20 mM glucose only) and positive (20 mM glucose plus PMA) controls (n = 3 experimental repeats) (Z'-factor = 0.69). All values represent mean ± SEM. *p<0.05, ***p<0.001.

secretion in response to acetylcholine through the muscarinic GPCRs, as well as the ionotropic nicotinic acetylcholine receptors (tyrothricin, tomatine, carbachol, and tacrine hydrochloride) (*Figure 2B and C*; *Changeux et al., 1969*; *Milner et al., 2011*; *Rosenberry et al., 2008*; *Marco and Carreiras, 2003*; *Lang and Staiger, 2016*; *Shih et al., 2009*). One compound has been shown to promote insulin secretion by inhibition of the mitochondrial protein tyrosine phosphatase PTPM1 (alexidine hydrochloride) (*Figure 2B and C*; *Doughty-Shenton et al., 2010*; *Nath et al., 2015*), and another likely affects β-cell function by signaling through the mineralocorticoid receptor (deoxycorticosterone) (*Figure 2B and C*; *Lu et al., 2006*). Finally, in addition to ion channels and GPCRs, the macrolide IVM has also been shown to signal in micromolar concentrations though several ionotropic receptors, including purinergic, GABAergic, and glycine receptors, as well as through the farnesoid X nuclear receptor (*Chen and Kubo, 2018*; *Dawson et al., 2000*; *Soltani et al., 2011*).

10 of these 12 hit compounds were not considered for further analysis because of either the high dose required to achieve insulin secretion (*Figure 2—figure supplement 1B*) or because they augmented insulin release in low basal glucose (2 mM) in intact WT mouse primary islets (*Figure 2D*). One of the remaining compounds induces hepatotoxicity after prolonged use (tacrine hydrochloride) (*Galisteo et al., 2000*). We therefore focused our attention on IVM due to its dose-dependent enhancement of GSIS in insulin-NanoLuc-expressing Beta-TC-6 cells, as well as its robust rescue of insulin secretion in *Bmal1*[-/-] islets (*Figure 2D and E*).

## Lead compound ivermectin regulates glucose-stimulated calcium flux and insulin exocytosis in *Bmal1* mutant islets

To test whether IVM drives GSIS in β-cell lines and primary mouse islets, we first assessed the impact of both acute treatment (1 hr) and overnight exposure (24 hr) with 10 µM IVM on the ability of WT β-cells and mouse islets to secrete insulin (*Figure 3A*, *Figure 3—figure supplement 1A*). Consistent with our initial bioluminescence assay, we observed that IVM enhanced insulin secretion in a glucose-dependent manner following both 1 hr IVM exposure and 24 hr pretreatment with IVM in β-cell lines and WT mouse islets, suggesting that both acute and longer-term exposure to IVM enhance β-cell function (*Figure 3A*, *Figure 3—figure supplement 1A*). Since there was not a significant increase in insulin secretion with overnight (approximately twofold) compared to acute (~1.5–1.6-fold) IVM exposure, further analysis of IVM as a potentiator of insulin secretion was performed only with acute treatment.

Chemical energy from ATP generated by glucose metabolism within the β-cell triggers closure of the sulfonylurea-linked potassium channel, depolarization of the plasma membrane, and opening of voltage-gated calcium channels, leading to stimulus-secretion coupling. To assess the mechanism of IVM-induced insulin secretion, we next monitored real-time calcium influx using ratiometric fluorescence imaging in WT β-cells in the presence of both glucose and IVM. We observed an immediate and robust glucose-stimulated intracellular calcium response within 2 min of IVM stimulation (p<0.05) (*Figure 3—figure supplement 1B*). Importantly, this effect was only observed in the presence of high glucose, consistent with results of our initial NanoLuc 384-well plate screening and subsequent ELISA-based analyses of GSIS. In contrast, the Ca$^{2+}$ channel inhibitor isradipine completely suppressed Ca$^{2+}$ influx and insulin secretion (*Figure 3—figure supplement 1C and D*; *Berjukow et al., 2000*). To determine whether increased calcium influx corresponded with productive insulin release following IVM treatment, we used a dynamic perifusion system to directly measure NanoLuc activity in eluates harvested from IVM-treated β-cells every 2 min over the course of 30 min following stimulation with either 20 mM glucose or 20 mM glucose plus 10 µM IVM (*Figure 3—figure supplement 1E*). IVM significantly increased insulin release during the initial burst of secretion within the first

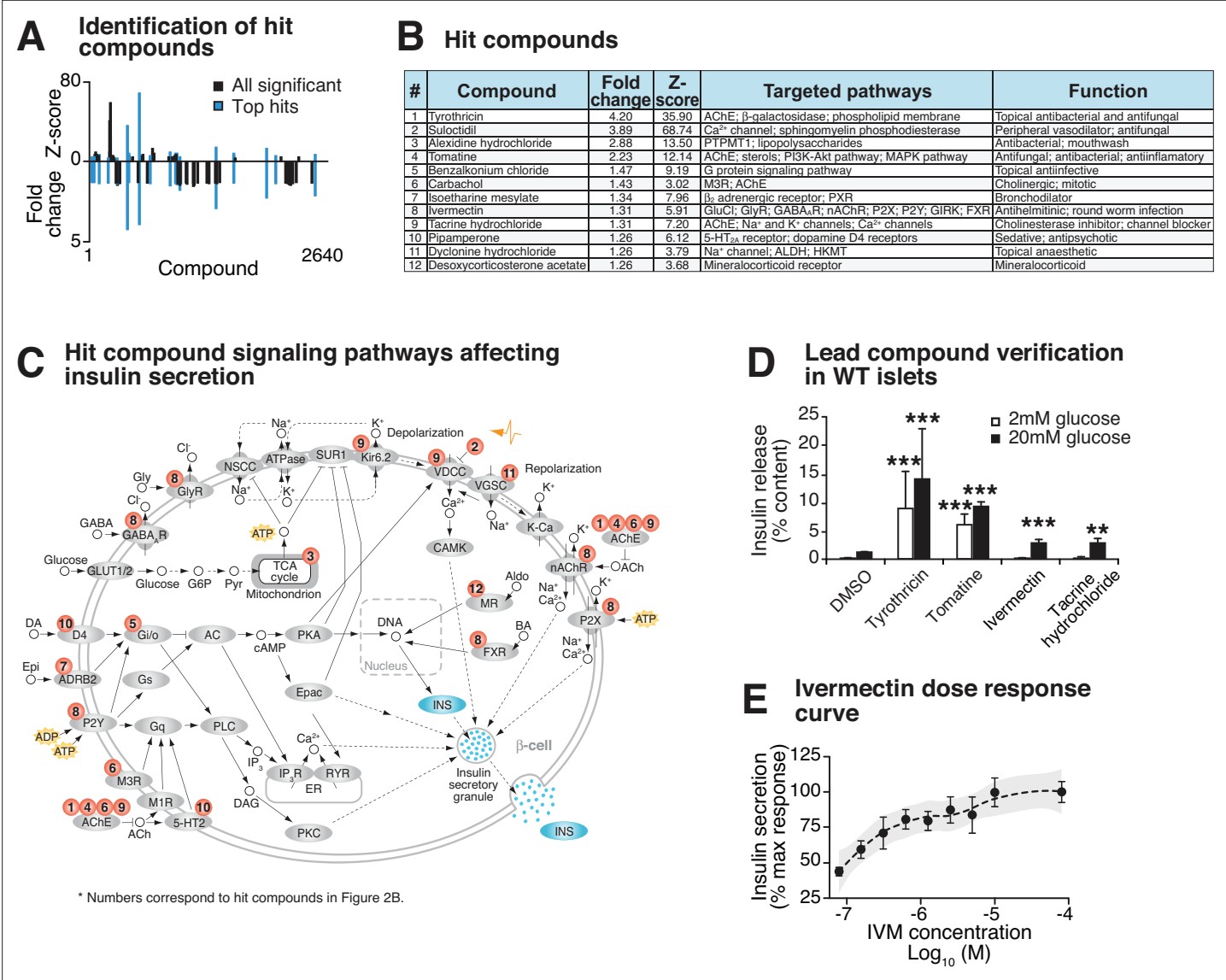

**Figure 2.** Identification and validation of high-throughput screen lead compounds in murine islets at high and low glucose concentrations. (**A**) Significant Z-scores (>3 standard deviations) and fold changes (>1.25-fold increase) for all 2640 screened compounds, with hit compounds indicated in blue. (**B**) Top 12 hit compounds identified from screen with a fold increase > 1.25 and a Z-score > 3, which were selected for further analysis. Known functions and published molecular pathways targeted by these compounds are indicated. (**C**) Model of potential mechanisms of action of the top 12 hit compounds to affect insulin secretion in the β-cell. (**D**) Glucose-responsive insulin secretion by ELISA at 2 mM and 20 mM glucose in WT mouse islets following exposure to four lead candidate compounds (n = 3–11 mice/compound). (**E**) Ivermectin (IVM) dose-response curve (n = 6–8 experimental repeats/dose), ranging from 0.078 μM to 80 μM IVM, in insulin-NanoLuciferase-expressing Beta-TC-6 cells. Shaded area represents 95% confidence intervals for the LOESS curve. All values represent mean ± SEM. **$p<0.01$, ***$p<0.001$.

The online version of this article includes the following figure supplement(s) for figure 2:

**Figure supplement 1.** High-throughput screen for modulators of insulin secretion in circadian mutant β-cells and validation of lead compounds.

12 min post-stimulation ($p<0.05$) and continued to enhance insulin secretion during the remainder of the stimulation period (12–30 min), consistent with continuous release of reserve insulin granules (*Rorsman and Renström, 2003*).

Since our cell-based studies indicated that IVM stimulates GSIS within immortalized β-cell lines, we next sought to determine whether IVM restores insulin secretion in the context of circadian disruption within primary islets, which are composed of multiple hormone-releasing cell types (*Arrojo E Drigo et al., 2020*). To test this idea, we administered IVM to mouse islets isolated from pancreas-specific *Bmal1*⁻/⁻ mice, revealing a 3.3-fold elevation of GSIS following exposure to the drug in the *Bmal1*

**A** Acute and overnight IVM treatments increase insulin secretion in WT islets

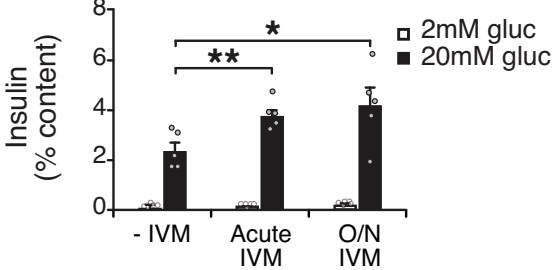

**D** IVM rescues insulin secretion defect in *Cry* double mutant islets

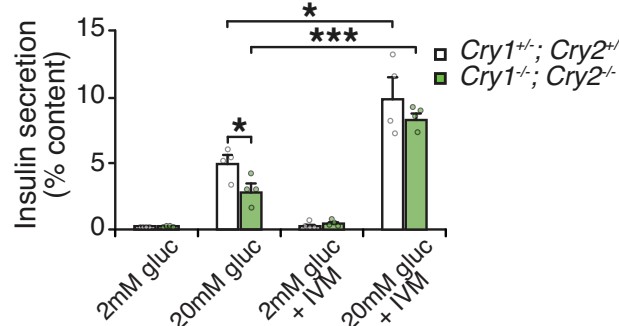

**B** IVM rescues insulin secretion defect in *Bmal1* mutant islets

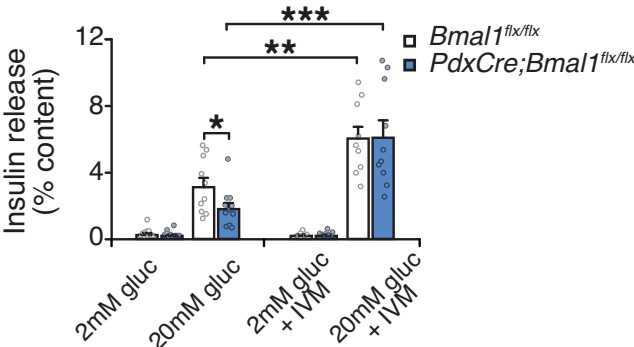

**E** IVM restores insulin exocytosis in *Bmal1* mutant islet β cells

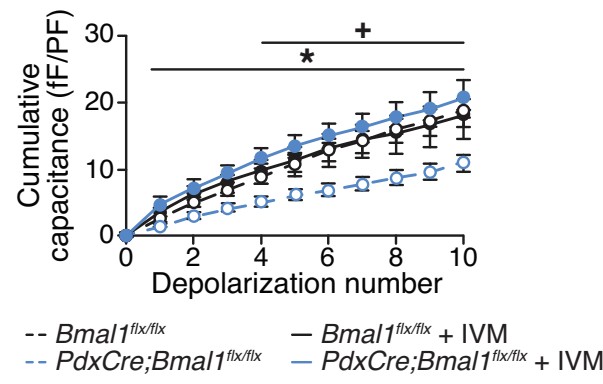

**C** IVM augments *Bmal1* mutant islet insulin release during perifusion

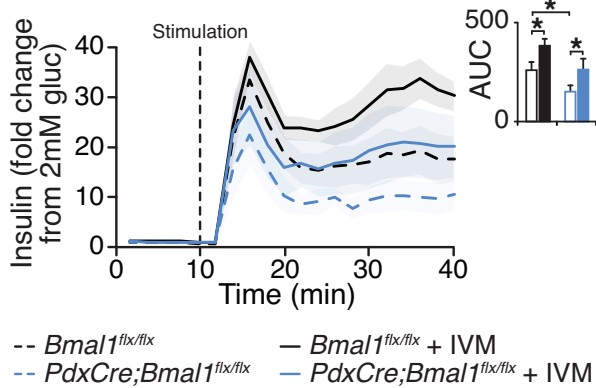

**F** IVM increases insulin exocytosis in human islet β cells

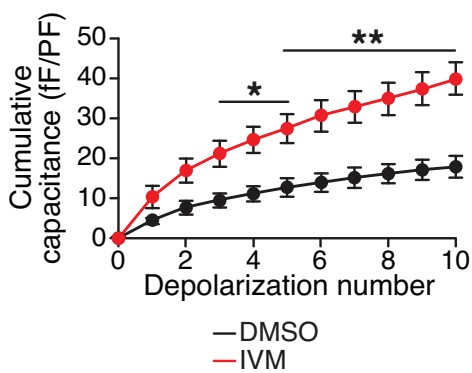

**Figure 3.** Effect of lead compound ivermectin (IVM) on glucose-stimulated insulin exocytosis and calcium flux from WT and circadian mutant β-cells. (**A**) Insulin secretion (expressed as % content) assessed by ELISA at 2 mM and 20 mM glucose in WT mouse islets in response to 1 hr 10 μM IVM treatment or 24 hr 10 μM IVM pretreatment (n = 5 mice). Data was analyzed by two-way ANOVA and false discovery rate (FDR) correction for multiple testing. (**B**) Insulin secretion as assessed by ELISA from islets isolated from 8-month-old pancreas-specific *Bmal1* knockout and *Bmal1^flx/flx^* mice in the

*Figure 3 continued on next page*

*Figure 3 continued*

presence or absence of 10 μM IVM (n = 10–11 mice/genotype). (**C**) Perifusion analysis of insulin secretion in islets from pancreas-specific *Bmal1* knockout (*PdxCre;Bmal1*<sup>flx/flx</sup>) and *Bmal1*<sup>flx/flx</sup> mice in response to 10 μM IVM in the presence of 20 mM glucose (n = 3 mice/genotype). (**D**) Insulin secretion as assessed by ELISA from islets isolated from 9- to 12-month-old male *Cry1*<sup>-/-</sup>;*Cry2*<sup>-/-</sup> knockout and *Cry1*<sup>+/-</sup>;*Cry2*<sup>+/-</sup> heterozygous control mice in the presence or absence of ± μM IVM (n = 4 mice/genotype). (**E**) Capacitance measurements in β-cells from *PdxCre;Bmal1*<sup>flx/flx</sup> and *Bmal1*<sup>flx/flx</sup> mouse islets treated with 10 μM IVM (n = 4–5 mice/genotype, 5–16 cells per mouse). Asterisks denote significance between *PdxCre;Bmal1*<sup>flx/flx</sup> and *PdxCre;Bmal1*<sup>flx/flx</sup> + IVM; plus symbols denote significance between *Bmal1*<sup>flx/flx</sup> and *PdxCre;Bmal1*<sup>flx/flx</sup> for all depolarization numbers indicated. */+ p<0.05. (**F**) Capacitance measurements in β-cells from human islets treated with 10 μM IVM (n = 3 donors, 7–11 cells per donor). Capacitance and calcium data were analyzed by two-way repeated-measures ANOVA with Bonferroni correction for multiple testing. All values represent mean ± SEM. *p<0.05, **p<0.01, ***p<0.001.

The online version of this article includes the following figure supplement(s) for figure 3:

**Figure supplement 1.** Ivermectin (IVM) improves insulin exocytosis in diabetic mice.

mutant islets (*Figure 3B*). Furthermore, perifusion experiments in islets from *Bmal1* mutant mice revealed that IVM significantly increased insulin release during both the initial burst of secretion (first 12 min post-stimulation) and during the sustained release (12–30 min) in both WT and *Bmal1* mutant islets (*Figure 3C*). Additionally, we observed a similar 2.9-fold increase in GSIS following administration of IVM to islets isolated from an independent mouse model of circadian disruption (*Cry1*<sup>-/-</sup>;*Cry2*<sup>-/-</sup> mice) (*Figure 3D*), suggesting that IVM ameliorates secretory defects caused by disruption of the circadian clock network. To determine if IVM can improve glucose homeostasis in diabetic animals, we next tested the effects of chronic IVM administration in the well-characterized C57BL/6-*Ins2*<sup>Akita</sup>/J *Akita* model of β-cell failure (*Yoshioka et al., 1997*). Daily intraperitoneal IVM (1.3 mg/kg body weight) was administered to *Akita* mice over a 14-day period (*Jin et al., 2013*), terminating in assessment of glucose tolerance and ex vivo GSIS. Treatment with IVM significantly improved glucose tolerance and augmented glucose-stimulated insulin release from islets isolated from these mice (*Figure 3—figure supplement 1F and G*). Given that our prior genomic and cell physiological studies have localized the β-cell defect in circadian mutant mice to impaired insulin exocytosis (*Marcheva et al., 2020*), and as IVM augmented insulin secretion in *Bmal1* mutant islets, we next sought to determine whether IVM might enhance depolarization-induced exocytosis using electrophysiological analyses (*Fu et al., 2019*). We assessed cumulative capacitance, a measure of increased cell surface area as insulin granules fuse to the plasma membrane, in β-cells from islets of control and pancreas-specific *Bmal1* mutant mice, as well as from human cadaveric islets. While *Bmal1* mutant cells displayed reduced rates of exocytosis following direct depolarization (as indicated by reduced capacitance), 10 μM IVM treatment rescued the defect in *Bmal1* mutant cells, increasing cumulative capacitance from 11.0 to 20.7 fF/pF after 10 consecutive depolarization steps (*Figure 3E*). IVM treatment also enhanced cumulative capacitance in human β-cells from 17.9 to 39.7 fF/pF (*Figure 3F*). Together, these data show that IVM augments β-cell early calcium influx in a glucose-dependent manner to promote increased vesicle fusion and release.

## Purinergic receptor P2Y1 mediates IVM-induced insulin exocytosis

Several of the predicted targets of the insulinotropic compounds from our screen, including IVM, involve second-messenger signaling, raising the possibility that circadian disruption may be overcome by augmenting hormonal or metabolic factors that promote peptide exocytosis. IVM is a readily absorbable and potent derivative of avermectin B$_1$ that allosterically regulates several different types of cell surface receptors, including purinergic and GABA receptors, as well as nuclear transcription factors such as the farnesoid X receptor (FXR) (*Jin et al., 2013*; *Khakh et al., 1999*; *González Canga et al., 2008*; *Estrada-Mondragon and Lynch, 2015*). Since IVM augments insulin secretion in *Bmal1*<sup>-/-</sup> cells, we hypothesized that the expression of putative IVM targets may be reduced during circadian disruption. We first identified the purinergic receptor P2Y1 (*P2ry1*) as the most highly expressed putative IVM target in wild-type β-cells (*Figure 4A*). We then observed that *P2ry1* was one of the most highly downregulated targets in *Bmal1*<sup>-/-</sup> cells, with mRNA expression levels reduced by ~3.1-fold (adjusted p=10<sup>-55</sup>; *Figure 4A*, *Figure 4—figure supplement 1A*; GSE146916). We found decreased levels and loss in rhythmicity of *P2ry1* in synchronized *Bmal1*<sup>-/-</sup> pseudoislets (*Figure 4—figure supplement 1B*). BMAL1 chromatin immunoprecipitation-sequencing in Beta-TC-6 cells further revealed enrichment of BMAL1 chromatin binding within enhancer regions 266–41 kb upstream of the *P2ry1* gene transcription start site (GSE69889; *Figure 4A*, *Figure 4—figure supplement 1A*). Finally,

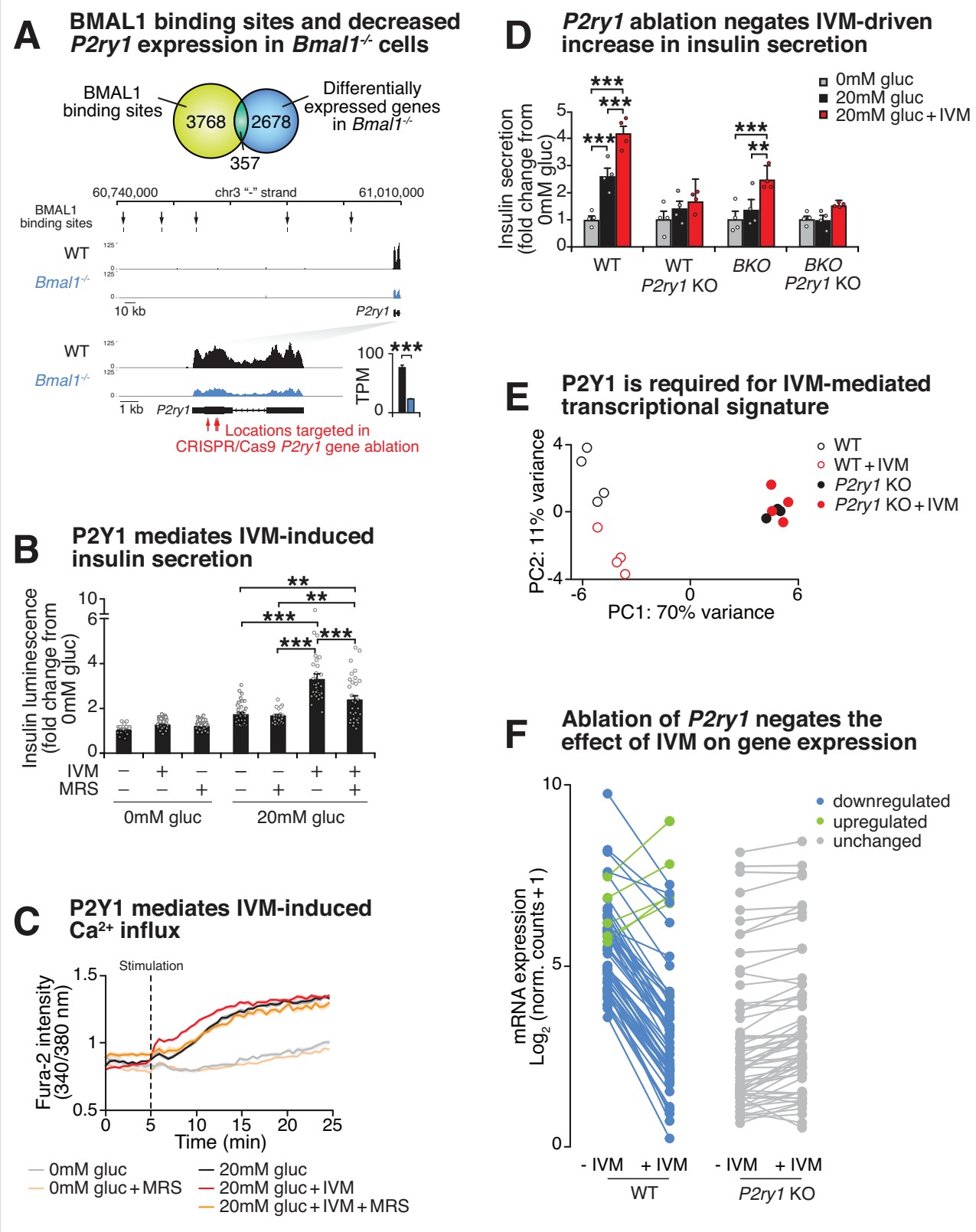

**Figure 4.** Purinergic receptor P2Y1 is required for ivermectin (IVM) to augment insulin exocytosis. (**A**) Venn diagram of BMAL1 binding sites identified by ChIP-sequencing overlapping with differentially expressed genes identified by RNA-sequencing in *Bmal1⁻/⁻* β-cell line compared to control cell line (top). Browser tracks and bar graph showing decreased expression of *P2ry1* gene in *Bmal1⁻/⁻* cells compared to controls. BMAL1 binding sites upstream of the *P2ry1* gene are also indicated (bottom). (**B**) Bioluminescence from WT insulin-NanoLuciferase pseudoislets in response to 10 μM IVM and/or 10 μM of

*Figure 4 continued on next page*

*Figure 4 continued*

the P2Y1 antagonist MRS2179 (n = 3–8 experiments, 3–15 experimental repeats/experiment). (**C**) Ratiometric determination of intracellular Ca$^{2+}$ using Fura2-AM dye in WT Beta-TC-6 cells stimulated in the presence or absence of 10 µM IVM (n = 3–7 experiments, 4–19 experimental repeats/experiment). (**D**) Insulin secretion by ELISA in pseudoislets from *P2ry1* KOs and control WT and *Bmal1*$^{-/-}$ Beta-TC-6 cells (n = 4 experiments, two experimental repeats/experiment). p-Values were determined by Tukey's multiple comparison tests following two-way ANOVA. (**E**) First two principal components (PC1 and PC2) following unbiased principal component analysis (PCA) of DESeq2 normalized counts in WT, WT + IVM, *P2yr1* KO, and *P2yr1* KO cells (n = 4 per group). (**F**) Mean log$_2$-transformed DESeq2-normalized counts in WT, WT + IVM, *P2yr1* KO, and *P2yr1* KO cells (n = 4 per group) at differentially expressed (1.5-fold, adjusted p-value<0.05) transcripts identified between WT and WT + IVM treated cells. All values represent mean ± SEM. *p<0.05, **p<0.01, ***p<0.001.

The online version of this article includes the following source data and figure supplement(s) for figure 4:

**Figure supplement 1.** Evidence for circadian control of *P2ry1*.

**Figure supplement 2.** Genetic ablation of purinergic receptor P2Y1 in Beta-TC-6 cells blunts effect of ivermectin (IVM) on gene expression.

**Figure supplement 2—source data 1.** P2Y1 expression by Western blot.

**Figure supplement 2—source data 2.** ACTIN expression by Western blot.

---

analysis of RNA-sequencing data from human islets (SRA accession ERP017126) indicates that *P2RY1* expression is enriched within β-cells among hormone-secreting cell types, with little to no detectable expression in the glucagon-secreting α cells (*Figure 4—figure supplement 1C*; *Segerstolpe et al., 2016*). Together, these data reveal direct rhythmic control of the *P2ry1* gene by the β-cell circadian clock.

Based upon evidence that IVM targets purinergic receptors (*Weng et al., 2008*; *Priel and Silber-berg, 2004*; *Bowler et al., 2003*; *Hansen et al., 2008*), that the predominant purinergic receptor on β-cells is P2Y1, and that BMAL1 specifically controls *P2ry1* amongst the purinergic receptor family in the β-cell (*Figure 4A*, *Figure 4—figure supplement 1A and B*), we sought to test the functional role of the P2Y1 receptor in the insulinotropic action of IVM. Pharmacological inhibition of P2Y1 using a subtype-specific inhibitor, the nucleotide analog MRS2179, in the presence of both high glucose and 10 µM IVM resulted in a 52% reduction in insulin secretion by bioluminescence and a reduction in calcium influx to levels similar to those observed during high glucose alone, as assessed by Fura2-AM ratiometric determination of intracellular calcium (*Figure 4B and C*). In addition to evidence that pharmacological blockade of P2Y1 receptor signaling attenuates IVM activity, we also tested the requirement of P2Y1 receptor signaling following CRISPR-Cas9-mediated knockout of the P2Y1 receptor in both WT and *Bmal1*$^{-/-}$ β-cells (*Figure 4—figure supplement 2A*). While IVM enhanced GSIS in WT and *Bmal1*$^{-/-}$ β-cells 1.6- and 1.8-fold, respectively, IVM did not significantly enhance GSIS in cells lacking the P2Y1 receptor (*Figure 4D*). Similar to the pharmacological findings with the P2Y1 antagonist MRS2179, these results demonstrate a requirement for P2Y1 in IVM-induced GSIS.

P2Y1 receptor signaling involves activation of Ca$^{2+}$ entry and intracellular release, which results in both acute stimulation of insulin granule trafficking and activation of transcription factors that may be involved in β-cell function (*Léon et al., 2005*; *Khan et al., 2014*; *Balasubramanian et al., 2010*). To analyze gene expression changes induced by P2Y1 activation, we performed RNA-sequencing to compare the IVM response within both WT and *P2ry1*$^{-/-}$ β-cells following stimulation with glucose or glucose plus IVM. Principal component analysis (PCA) was performed using log-transformed count data from the top 500 most variable genes across all samples (*Love et al., 2014*). This revealed distinct patterns in mRNA expression between IVM- and control-treated WT cells along PC2, while there was no separation between IVM- and control-treated *P2ry1*$^{-/-}$ β-cells, suggesting that P2Y1 is required for IVM-mediated transcriptional changes in β-cells (*Figure 4E*). In WT cells, IVM induced differential expression of 65 transcripts (1.5-fold change, adjusted p-value<0.05), including upregulation of the immediate early gene *Fos* (*Murphy et al., 1991*) and downregulation of *Aldolase B*, whose expression has been linked to reduced insulin secretion in human islets (*Gerst et al., 2018*; *Figure 4F*, *Figure 4—figure supplement 2B*, *Supplementary file 2*). Strikingly, none of these transcripts were significantly altered by IVM in the *P2ry1*$^{-/-}$ β-cells (all adjusted p-value>0.05) (*Figure 4F*, *Figure 4—figure supplement 2B*, *Supplementary file 2*). Taken together, these data suggest that the circadian clock program controls P2Y1 expression to modulate GSIS and highlight the utility of a genetic-sensitized drug screen for identification of therapeutic targets in circadian dysregulation and diabetes.

## Discussion

We have identified an unexpected role for the P2Y1 receptor as a BMAL1-controlled insulinotropic factor required for enhanced β-cell glucose-stimulated $Ca^{2+}$ influx and insulin secretion in response to IVM. While P2Y receptors have been previously implicated in calcium and insulin secretory dynamics in β-cells, modulation has been primarily demonstrated using agonists that mimic ATP/ADP derivatives that have deleterious effects on thrombosis (*Léon et al., 2005*; *Khan et al., 2014*; *Balasubramanian et al., 2010*; *Gąsecka et al., 2020*). Little is known about P2Y1 targeting in disease states, such as circadian disruption and/or type 2 diabetes, or whether P2Y1 is controlled at a transcriptional level. Our evidence that P2Y1 is expressed under control of the circadian clock derives from analyses at the level of both chromatin binding by the core clock factor BMAL1 and genome-wide differential RNA expression analysis in circadian mutants. Intriguingly, P2X and P2Y receptors are required for $Ca^{2+}$ signaling in the suprachiasmatic nucleus (*Lommen et al., 2017*; *Svobodova et al., 2018*), yet their role in circadian regulation of peripheral tissues has not been well studied. Our data suggests that IVM action requires the presence of P2Y1 receptors in β-cells since functional ablation of the P2Y1 receptor attenuates the effect of IVM on insulin secretion in both wild-type and circadian mutant β-cells (*Figure 4D*). Our analyses reveal that pharmacological enhancement of P2Y1 receptor activity may therefore bypass pathological and circadian alterations in expression of the P2Y1 receptor in β-cells to restore insulin secretion. Recently, the P2Y1 receptor was implicated in nutrient- and ATP/ADP-dependent regulation of insulin release through an adipocyte-islet axis, further suggesting that P2RY1 may play a role in physiological regulation of islet hormone release (*Prentice et al., 2021*). Future studies will be required to determine whether IVM affects paracrine ATP/ADP release to affect P2RY1 or whether IVM directly binds purinergic receptors in the β-cell. One possibility is that IVM may augment P2X-P2Y1 crosstalk to drive insulin secretion, which has been shown to drive $Ca^{2+}$ and P2Y1-dependent activation of other cell types (*Weng et al., 2008*; *Woehrle et al., 2019*).

Previous physiological and transcriptomic studies have shown that circadian regulation of insulin exocytosis involves control of the expression and activity of cell-surface receptors and second messenger systems (*Perelis et al., 2015*; *Gil-Lozano et al., 2014*). We based our drug screen on the idea that modulators of insulin secretion in cells that lack a functional clock would complement prior genomic analyses revealing circadian control of peptidergic hormone exocytosis and also to provide proof of principle that the clock can be leveraged to sensitize screening for new chemical modulators of β-cell function. This approach identified $Ca^{2+}$-dependent pathways as a potential route to ameliorate circadian disruption and enhance GSIS. Previous small-molecule screens have identified glucose-dependent insulinotropic compounds in wild-type cells (*Burns et al., 2015*). However, several of these compounds, including the anti-inflammatory bufexamac and anti-giardiasis drug lobendazole, were found to be ineffective or even inhibitory in our circadian mutant screen (*Supplementary file 1*). In the future, high-throughput screens may lead to more personalized therapeutics through comparison of insulinotropic compounds identified using cells without known mutations versus those discovered in cells harboring monogenic or polygenic diabetes variants. Several of the compounds identified in our screen have been used in disease treatment and have known mechanisms of action, including the cholinergic activators carbachol and tacrine (*Linn et al., 1989*; *Crismon, 1994*). The identification of these compounds in our screen raises the intriguing possibility of using drug derivatives related to these molecules for type 2 diabetes treatment, particularly in the context of circadian/sleep disruption.

The study of transcriptional rhythms across the 24 hr circadian cycle has previously revealed a diverse landscape of clock-controlled genes and pathways (*Zhang et al., 2014*). Despite the identification of thousands of tissue-specific and clock-controlled transcripts, limited advances have been made in utilizing this information to treat diseases associated with circadian disruption, including type 2 diabetes. One approach to this challenge has been to intervene and restore the molecular clock program using pharmacology (Nobiletin) (*He et al., 2016*), micronutrient supplementation ($NAD^+$ precursors) (*Levine et al., 2020*; *Sato et al., 2017*), or enforced behavioral rhythms (such as time-restricted feeding) (*Sutton et al., 2018*). However, it remains unclear how altering the whole-body clock will affect nutritional and hormonal dynamics at a cellular level. Another approach has been to directly target clock-controlled genes with known function in health and disease (*Lamia et al., 2008*) or to look at gain/loss of circadian control in health versus disease (*Petrenko et al., 2020a*). This approach requires an understanding of gene function within a given tissue, and thus limits the identification of novel therapeutic targets. In the studies performed here, we sought to address the

challenge of connecting clock control of transcription with druggable targets by using an unbiased small-molecule drug screen, in tandem with functional genomics, to elucidate mechanisms of insulin secretory dynamics. Since the circadian timing system has been shown to not only regulate the function of mature β-cells, but also the regenerative capacity of islets in both the context of the mouse (*Petrenko et al., 2020b*) and in human embryonic stem cell differentiation (*Alvarez-Dominguez et al., 2020*), molecules identified in cell-based genetic screens may provide broad applicability as therapeutics.

# Materials and methods

## Key resources table

| Reagent type (species) or resource | Designation | Source or reference | Identifiers | Additional information |
|---|---|---|---|---|
| Gene (*Mus musculus*) | *Bmal1* | NCBI | Gene ID: 11865 | |
| Gene (*M. musculus*) | *Cry1* | NCBI | Gene ID: 12952 | |
| Gene (*M. musculus*) | *Cry2* | NCBI | Gene ID: 12953 | |
| Gene (*M. musculus*) | *P2ry1* | NCBI | Gene ID: 18441 | |
| Strain (*M. musculus*), strain background (C57BL6J) | *PdxCre;Bmal1^{flx/flx}* | PMID:20562852 | *PdxCre;Bmal1^{flx/flx}* | Pancreas-specific *Bmal1* mutant mice |
| Strain (*M. musculus*), strain background (C57BL6J) | *Cry1^{-/-};Cry2^{-/-}* | PMID:10518585 | *Cry1^{-/-};Cry2^{-/-}* | Whole-body *Cry1* and *Cry2* double knockout mice |
| Strain (*M. musculus*), strain background (C57BL6J) | C57BL/6-*Ins2^{Akita}*/J | Jackson Laboratory | 3548 | Spontaneous mutation in the insulin two gene leads to incorrect folding of the insulin protein Monogenic mouse model for type 1 diabetes. |
| Biological sample (*M. musculus*) | Primary pancreatic islets | Northwestern University | | Freshly isolated from mice |
| Biological sample (*Homo sapiens*) | Primary pancreatic islets | Alberta Diabetes Institute Islet-Core | https://www.isletcore.ca (R224, R225, R226) | Freshly isolated from nondiabetic donors |
| Cell line (*H. sapiens*) | HEK293T | ATCC | CRL-11268 | Kidney cells |
| Cell line (*M. musculus*) | Beta-TC-6 | ATCC | CRL-11506 | Pancreatic β-cells |
| Cell line (*M. musculus*) | *Bmal1^{-/-}* Beta-TC-6 | PMID:21686102 | *Bmal1^{-/-}* Beta-TC-6 | Pancreatic β-cells, mutant for *Bmal1* |
| Cell line (*M. musculus*) | Insulin-NanoLuc-expressing Beta-TC-6 | This paper | | Pancreatic β-cells, expressing Proinsulin-NanoLuc luminescent reporter |
| Cell line (*M. musculus*) | Insulin-NanoLuc-expressing *Bmal1^{-/-}* Beta-TC-6 | This paper | | Pancreatic β-cells, mutant for *Bmal1*, expressing Proinsulin-NanoLuc luminescent reporter |
| Recombinant DNA reagent | P2Y1 CRISPR/Cas9 KO plasmid | Santa Cruz Biotechnology | sc-422095 | Pool of three plasmids, encoding the Cas9 nuclease and a P2Y1-specific 20 nt guide RNA, targeting exon 1 of the mouse *P2ry1* gene |
| Recombinant DNA reagent | P2Y1 HDR plasmid | Santa Cruz Biotechnology | sc-422095-HDR | Pool of 2–3 plasmids, containing puromycin resistance gene and HDR templates, corresponding to the P2Y1 CRISPR/Cas9 KO plasmid cut sites |
| Recombinant DNA reagent | Proinsulin-NanoLuc in pLX304 lentivirus packaging plasmid | Addgene | 62057 | Luminescent reporter of insulin secretion, mouse synthetic Ins2 insert |
| Recombinant DNA reagent | pCMV-VSV-G | Addgene | 8454 | Envelope vector |
| Recombinant DNA reagent | pCMV delta R8.2 | Addgene | 12263 | Packaging vector |
| Chemical compound, drug | Spectrum Collection | MicroSource Discovery Systems, Inc | | Small-molecule compound library, which consists of 2640 known drugs and drug-like molecules |
| Chemical compound, drug | Ivermectin | Tocris | 1260 | |
| Chemical compound, drug | (+)-Bicuculline | Tocris | 130 | |
| Chemical compound, drug | MRS2179 tetrasodium salt | Tocris | 900 | |
| Chemical compound, drug | Isradipine | Cayman Chemical Company | 17536 | |
| Chemical compound, drug | Phorbol 12-myristate 13-acetate (PMA) | Sigma-Aldrich | P1585 | |
| Chemical compound, drug | Carbamoylcholine chloride | Sigma-Aldrich | C4382-1G | |
| Chemical compound, drug | Forskolin | Sigma-Aldrich | F3917-10MG | |
| Chemical compound, drug | D-Glucose | Sigma-Aldrich | G7528-250G | |
| Chemical compound, drug | Potassium chloride | Sigma-Aldrich | P-5405 | |

*Continued on next page*

*Continued*

| Reagent type (species) or resource | Designation | Source or reference | Identifiers | Additional information |
|---|---|---|---|---|
| Chemical compound, drug | 3-Isobutyl-1-methylxanthine | Sigma-Aldrich | I7018-250mg | |
| Chemical compound, drug | Tyrothricin | Sigma-Aldrich | T3000000 | |
| Chemical compound, drug | Alexidine hydrochloride | Cayman Chemical Company | 13876 | |
| Chemical compound, drug | Benzalkonium chloride | Sigma-Aldrich | 12060-5G | |
| Chemical compound, drug | Suloctidil | MicroSource Discovery Systems, Inc | 01501153 | |
| Chemical compound, drug | Tomatine | MicroSource Discovery Systems, Inc | 01504079 | |
| Chemical compound, drug | Isoetharine mesylate | MicroSource Discovery Systems, Inc | 01505977 | |
| Chemical compound, drug | Tacrine hydrochloride | MicroSource Discovery Systems, Inc | 02300104 | |
| Chemical compound, drug | Pipamperone | MicroSource Discovery Systems, Inc | 01505690 | |
| Chemical compound, drug | Dyclonine hydrochloride | MicroSource Discovery Systems, Inc | 01500268 | |
| Chemical compound, drug | Desoxycorticosterone acetate | MicroSource Discovery Systems, Inc | 00300029 | |
| Chemical compound, drug | Puromycin dihydrochloride | Sigma-Aldrich | P8833-25MG | |
| Chemical compound, drug | Collagenase P | Sigma-Aldrich | C7657-100mg | |
| Chemical compound, drug | Biocoll | Millipore | L6155 | |
| Chemical compound, drug | 2-Hydroxypropyl-b-cyclodextrin | Sigma-Aldrich | H107-5G | |
| Chemical compound, drug | Fura-2 | Invitrogen | F1201 | |
| Chemical compound, drug | Pluronic F-127 | Invitrogen | P3000MP | |
| Chemical compound, drug | Penicillin-streptomycin | Gibco | 15-140-122 | |
| Chemical compound, drug | L-glutamine | Gibco | 25-030-081 | |
| Commercial assay or kit | Lipofectamine 3000 | Thermo Fisher Scientific | L3000015 | |
| Commercial assay or kit | NanoGlo Luciferase Assay Substrate | Promega | N1110 | |
| Commercial assay or kit | Ultra Sensitive Mouse Insulin ELISA Kit | Crystal Chem Inc | 90080 | |
| Commercial assay or kit | Autokit Glucose | Wako-Fujifilm | 997-03001 | |
| Commercial assay or kit | Direct-zol RNA Microprep kit | Zymo Research | R2062 | |
| Commercial assay or kit | High Capacity cDNA Reverse Transcription Kit | Applied Biosystems | 4368813 | |
| Commercial assay or kit | iTaq Universal SYBR Green Supermix | Applied Biosystems | 1725125 | |
| Commercial assay or kit | NEBNext Ultra Directional RNA Library Prep Kit for Illumina | New England Biolabs | E7760L | |
| Commercial assay or kit | NEBNext Library Quant Kit for Illumina | New England Biolabs | E7630L | |
| Commercial assay or kit | Quick Start Bradford Protein Assay | Bio-Rad | 500-0116 | |
| Sequence-based reagent | *β-actin* F | This paper | PCR primers | 5'-TGCTCTGGCTCCTAGCACCATGAAGATCAA-3' |
| Sequence-based reagent | *β-actin* R | This paper | PCR primers | 5'-AAACGCAGCTCAGTAACAGTCCGCCTAGAA-3' |
| Sequence-based reagent | *P2ry1* F | This paper | PCR primers | 5'-TTATGTCAGCGTGCTGGTGT-3' |
| Sequence-based reagent | *P2ry1* R | This paper | PCR primers | 5'-ACGTGGTGTCATAGCAGGTG-3' |
| Antibody | Anti-P2Y1, mouse monoclonal | Santa Cruz | sc-377324 | WB (1:500) |
| Antibody | Anti-β-ACTIN, rabbit monoclonal | Cell Signaling | CST 4970 | WB (1:4000) |
| Software, algorithm | STAR | PMID:23104886 | RRID:SCR_004463 | |
| Software, algorithm | RSEM | PMID:21816040 | RRID:SCR_013027 | |
| Software, algorithm | DESeq2 package in R | PMID:25516281 | RRID:SCR_015687 | |
| Software, algorithm | Pheatmap package in R | | RRID:SCR_016418 | |
| Software, algorithm | GraphPad Prism | GraphPad | RRID:SCR_002798 | |
| Software, algorithm | SRA-Toolkit | https://trace.ncbi.nlm.nih.gov/Traces/sra/sra.cgi?view=software | | |
| Software, algorithm | Seurat in package in R | PMID:34062119 | RRID:SCR_007322 | |

*Continued on next page*

*Continued*

| Reagent type (species) or resource | Designation | Source or reference | Identifiers | Additional information |
|---|---|---|---|---|
| Software, algorithm | JTK_Cycle | PMID:20876817 | RRID:SCR_017962 | |
| Other | Dulbecco's modified Eagle's medium | Gibco | 90-013-pb | |
| Other | Fetal bovine serum | Bio-Techne | S11550 | |
| Other | RIPA buffer | Sigma-Aldrich | R0278-50ML | |
| Other | Tri Reagent | Molecular Research Center, Inc | NC9277980 | |
| Other | Complete Mini EDTA-Free Protease Inhibitor | Roche | 4693159001 | |
| Other | PhoStop | Roche | 4906837001 | |
| Other | Nitrocellulose membranes | Bio-Rad | 1620112 | |

## Reagents

IVM, (+)-bicuculline, and MRS2179 tetrasodium salt were obtained from Tocris (R&D Systems, Inc, Minneapolis, MN). Isradipine and alexidine hydrochloride were purchased from Cayman Chemical Company (Ann Arbor, MI). PMA, carbamoylcholine chloride (carbachol), forskolin, tyrothricin, and benzalkonium chloride were obtained from Sigma-Aldrich (St. Louis, MO). Suloctidil, tomatine, isoetharine mesylate, tacrine hydrochloride, pipamperone, dyclonine hydrochloride, and desoxycorti-costerone acetate were purchased from MicroSource Discovery Systems, Inc.

## Animals

Male WT C57BL6J mice and C57BL/6-*Ins2*$^{Akita}$/J mice were purchased from the Jackson Laboratory (Bar Harbor, ME). *PdxCre;Bmal1*$^{flx/flx}$ and *Cry1*$^{-/-}$*;Cry2*$^{-/-}$ mice were produced and maintained on C57BL6J background at the Northwestern University Center for Comparative Medicine (Protocols IS00000466, IS00003253, IS00008732, IS0005838) (*Peek et al., 2013*; *Vitaterna et al., 1999*). Unless otherwise stated, animals were maintained on a 12:12 light:dark cycle and allowed free access to water and regular chow. All animal care and use procedures were conducted in accordance with regulation of the Institutional Animal Care and Use Committee at Northwestern University.

## Cell culture

Beta-TC-6 cells were obtained from ATCC (Manassas, VA) (CRL-11506), and *Bmal1*$^{-/-}$ Beta-TC-6 β-cell lines were previously derived as described (*Marcheva et al., 2020*). Cells were cultured in Dulbecco's modified Eagle's medium (DMEM; Gibco, Aramillo, TX) supplemented with 15% fetal bovine serum (Bio-Techne, Minneapolis, MN), 1% penicillin-streptomycin (Gibco), and 1% L-glutamine (Gibco) at 37°C with 5% CO$_2$. Culture medium was exchanged every 2–3 days. All cells used in experiments were at <15 passages. Cells were routinely checked for mycoplasma contamination.

## Generation of WT and *Bmal1*-/- Beta-TC-6 cells stably expressing insulin-NanoLuc

We used the proinsulin-NanoLuc plasmid (David Altshuler, Addgene plasmid #62057, proinsulin-NanoLuc in pLX304) to provide a low-cost, scalable, and rapid method to detect insulin secretion. The gene encoding NanoLuc was cloned into the C-peptide portion of mouse proinsulin such that cleavage within insulin vesicles by pH-sensitive prohormone convertase results in the co-secretion of NanoLuc with endogenous insulin in a stimulus-dependent manner (*Burns et al., 2015*). The pLX304 lentivirus packaging plasmid containing the proinsulin-NanoLuc construct was transfected into HEK293T (ATCC CRL-11268) cells with pCMV-VSVG (envelope vector, Addgene plasmid #8454) and pCMV delta R8.2 (packaging vector, Addgene plasmid #12263). Supernatant containing lentivirus particles was harvested 48 hr after transfection. Beta-TC-6 and *Bmal1*$^{-/-}$ Beta-TC-6 cells were infected with insulin-NanoLuc lentivirus, and stably expressing cells were selected by treating with puromycin (2 μg/ml, 2 days).

## CRISPR-mediated *P2ry1* deletion in WT and *Bmal1*-/- Beta-TC-6 cells

Exon 1 of the mouse *P2yr1* gene was deleted in WT and *Bmal1*$^{-/-}$ Beta-TC-6 cells by CRISPR-Cas9 and homology-directed repair (HDR). Cells were co-transfected with guide RNA, P2Y1 CRISPR/Cas9 KO,

and P2Y1 HDR plasmids (Santa Cruz Biotechnology, Dallas, TX, sc-422095 and sc-422095-HDR) by Lipofectamine 3000 (Thermo Fisher Scientific, Amarillo, TX). The locations of the three sites targeted for ablation by the P2Y1 CRISPR/Cas9 KO plasmids are indicated in *Figure 4A*. After 48 hr of transfection, stably integrated clones were selected for puromycin resistance (puromycin dihydrochloride, Sigma-Aldrich). RNA and protein were extracted from these colonies, and *P2ry1* expression was assessed by qPCR and Western blot.

## High-throughput screen for drugs to restore insulin secretion in *Bmal1-/-* β-cells and insulin secretion assays

The Spectrum Collection small-molecule compound library (MicroSource Discovery Systems, Inc), which consists of 2640 known drugs and drug-like molecules, was screened for compounds that augment insulin secretion in *Bmal1*$^{-/-}$ Beta-TC-6 cells. Insulin-NanoLuc-expressing *Bmal1*$^{-/-}$ Beta-TC-6 cells (40,000 cells/well) were placed into 384-well plates and cultured for 3 days at 37°C and 5% $CO_2$. The cells were washed once and incubated in KRB buffer containing 0 mM glucose for 1 hr. Then, KRB buffer containing 20 mM glucose in addition to the small molecules (10 μM) were added, and the cells were incubated for 1 hr. As a negative control, 16 wells received KRB buffer with only 20 mM glucose, which fails to elicit appropriate insulin secretion in *Bmal1*$^{-/-}$ cells, and as a positive control, 16 wells received KRB buffer containing 20 mM glucose and 10 μM PMA, which is known to induce insulin secretion in both *Bmal1*$^{-/-}$ mouse islets and Beta-TC-6 cells (*Perelis et al., 2015*). After 1 hr, the supernatant was collected and centrifuged at 500 × *g* for 30 min. The supernatant was transferred into a fresh 384-well assay plate containing NanoGlo Luciferase Assay Substrate (Promega, Madison, WI), and luciferase intensity was measured by EnSpire Plate Reader (PerkinElmer, Waltham, MA) within 30 min. All liquids for the high-throughput screen were dispensed using Tecan Fluent Automated Liquid Handling Platform (Tecan, Mannedorf, Switzerland) at the High-Throughput Analysis Laboratory at Northwestern University. Screen feasibility was determined by calculating Z'-factor using the following formula: Z'-factor = $1 - 3(\sigma_p + \sigma_n)/(\mu_p - \mu_n)$ (where $\sigma_p$ is the standard deviation of positive control [20 mM glucose + PMA], $\sigma_n$ is the standard deviation of negative control [20 mM glucose only], $\mu_p$ is the mean intensity of positive control, and $\mu_n$ is the mean intensity of the negative control) (*Zhang et al., 1999*).

## Determination of hit compounds

Z-scores are a measure of how many standard deviations above or below the population mean a raw score is. Z-scores for luciferase intensities produced by screened compounds were calculated from the following formula: $z = (X - \mu)/\sigma$ (where z is the Z-score, X is the luciferase intensity of the compounds, μ is the intensity of negative control [20 mM glucose], and σ is the standard deviation of negative control). A row-based correction factor was applied to all luciferase readings to adjust for logarithmic signal decay. Hit compounds were defined as those that elicited a response of greater than 3 standard deviations from the mean (Z-score > 3) and more than 1.25-fold increase compared to negative control, which is the cutoff for ~10% chance of the observation occurring by random chance. Validated hit compounds that augmented insulin secretion at low drug dose were considered lead compounds.

## Insulin secretion assays in pancreatic islets, pseudoislets, and cell lines

Mouse pancreatic islets were isolated via bile duct collagenase digestion (*Collagenase P*, Sigma) and Biocoll (Millipore) gradient separation and left to recover overnight at 37°C in RPMI 1640 with 10% FBS, 1% L-glutamine, and 1% penicillin/streptomycin. For insulin release assays, duplicates of five equally sized islets per mouse were statically incubated in Krebs-Ringer Buffer (KRB) at 2 mM glucose for 1 hr and then stimulated for 1 hr at 37°C with 2 mM or 20 mM glucose in the presence or absence of 10 μM of each compound. Supernatant was collected and assayed for insulin content by ELISA (Crystal Chem Inc, Elk Grove Village, IL). Islets were then sonicated in acid-ethanol solution and solubilized overnight at 4°C before assaying total insulin content by ELISA. For insulin release assays from pseudoislets, 3 × 10$^6$ cells were plated for 3 days in 60 mm suspension dishes and allowed to form pseudoislets for 2–3 days. Glucose-responsive insulin secretion was performed as described above using 10 pseudoislets per sample and a basal glucose level of 0 mM glucose instead of 2 mM (*Marcheva et al., 2020*). For secretion from insulin-NanoLuc cell lines, 1 × 10$^5$ cells were cultured on poly-L-lysine-coated 96 well plates for 2–3 days, starved for 1 hr in 0 mM glucose KRB, then

stimulated with indicated compounds and/or receptor antagonists for 1 hr in conjunction with 0 mM basal glucose or 20 mM stimulatory glucose conditions. Luciferase intensity after addition of NanoGlo to supernatant was measured by Cytation3 Plate Reader (BioTek, Winooski, VT).

## Perifusion of primary islets and pseudoislets

Primary islets from *PdxCre;Bmal1*<sup>flx/flx</sup> and *Bmal1*<sup>flx/flx</sup> mice were isolated as described above and left to recover overnight. Perifusion of 100 islets per mouse per treatment was performed using a Biorep Technologies Perifusion System Model PERI-4.2 with a rate of 100 µl/min KRB (0.1% BSA). After 1 hr of preincubation and equilibration at a rate of 100 µl/min with 2 mM glucose KRB, islets were perifused for 10 min with 2 mM glucose KRB, followed by perifusion for 30 min with 20 mM glucose or 20 mM glucose plus IVM. Perifusate was collected in 96-well plates, and insulin secreted was analyzed via ELISA. Perifusion of insulin-NanoLuc pseudoislets was performed in an identical manner using 0 mM glucose KRB instead of 2 mM glucose KRB. Pseudoislet perifusate was analyzed for NanoLuc activity using NanoGlo Luciferase Assay Substrate (Promega) as per the manual instructions.

## In vivo ivermectin treatment and glucose measurements

Mice were injected intraperitoneally for 14 days with 1.3 mg/kg body weight of IVM, which was dissolved in 40% w/v 2-hydroxypropyl-β-cyclodextrin (Sigma-Aldrich) (*Jin et al., 2013*). At the end of IVM treatment, mice were fasted for 14 hr and glucose tolerance tests were performed at ZT2 following intraperitoneal glucose injection at 2 g/kg body weight. Plasma glucose levels were measured by enzymatic assay (Autokit Glucose, Wako-Fujifilm, Cincinnati, OH).

## Synchronization, RNA isolation, and qPCR mRNA quantification

Where indicated, circadian synchronization was performed using 200 WT pseudoislets by first exposing cells to 10 µM forskolin for 1 hr, followed by transfer to normal media and RNA collection every 4 hr 24–44 hr following forskolin synchronization pulse. RNA was extracted from Beta-TC-6 cells and pseudoislets using Tri Reagent (Molecular Research Center, Inc, Cincinnati, OH) and frozen at −80°C. RNA was purified according to the manufacturer's protocol using the Direct-zol RNA Microprep kit (Zymo Research, Irvine, CA) with DNase digestion. cDNAs were then synthesized using the High Capacity cDNA Reverse Transcription Kit (Applied Biosystems, Amarillo, TX). Quantitative real-time PCR analysis was performed with SYBR Green Master Mix (Applied Biosystems) and analyzed using a Touch CFX384 Real-Time PCR Detection System (Bio-Rad, Hercules, CA). Target gene expression levels were normalized to *β-actin* and set relative to control conditions using the comparative $C_T$ method. Primer sequences for qPCR are as follows: *β-actin* forward: 5'-TGCTCTGGCTCCTAGCACCATGAAGATCAA-3', reverse: 5'-AAACGCAGCTCAGTAACAGTCCGCCTAGAA-3'; *P2ry1* forward: 5'- TTATGTCAGCGT-GCTGGTGT-3', reverse: 5'-ACGTGGTGTCATAGCAGGTG-3'.

## RNA-sequencing and analysis

Following RNA isolation (described above), RNA quality was assessed using a Bioanalyzer (Agilent, Santa Clara, CA), and sequencing libraries were constructed using a NEBNext Ultra Directional RNA Library Prep Kit for Illumina (New England Biolabs, Ipswich, MA, E7420L) according to the manufacturer's instructions. Libraries were quantified using a NEBNext Library Quant Kit for Illumina (New England Biolabs, E7630L) and sequenced on an Illumina NextSeq 500 instrument using 42 bp paired-end reads. For differential expression analysis, RNA raw sequence reads were aligned to the reference genome (mm10) using STAR version 2.7.2b, and raw and transcripts per million (TPM) count values determined using RSEM version 1.3.3. Differentially expressed RNAs were identified by a false discovery rate (FDR)-adjusted p-value<0.05 and a fold change > 1.5 using DESeq2 version 1.32.0 in R 4.1.0. Heatmaps were generated using the pheatmap package in R. Raw mRNA sequencing data and gene abundance measurements have been deposited in the Gene Expression Omnibus under accession GSE186469.

## Intracellular calcium determination

Beta-TC-6 cells were plated at a density of 100,000 cells per well in black 96-well plates with clear bottoms and cultured overnight at 37°C and 5% $CO_2$. Cells were then washed with BSA-free KRB buffer with no glucose and loaded with 5 µM Fura-2 (Invitrogen, Amarillo, TX) and 0.04% Pluronic

F-127 (Invitrogen) for 30 min at 37°C. Following a wash with BSA-free KRB, Fura-2 intensity was measured after stimulation with either glucose alone or glucose plus the indicated compounds. Cells were alternately excited with 340 nm and 380 nm wavelength light, and the emitted light was detected at 510 nm using a Cytation 3 Cell Imaging Multi-Mode Reader (BioTek) at sequential 30 s intervals. Raw fluorescence data were exported to Microsoft Excel and expressed as the 340/380 ratio for each well.

## Human islet studies and ethics statement

Human islet isolations and human islet cell biology experiments approved by the University of Alberta Human Research Ethics Board (approval identifiers: Pro00013094; Pro00001754) were performed at the Alberta Diabetes Institute Islet-Core according to the methods deposited in the protocols. io repository (*Isolation of Human Pancreatic Islets of Langerhans for Research V.3, 2021*). Organ donation was coordinated by the appropriate regional organ procurement organization, including obtaining written next-of-kin consent for use of donor organs in this study. Donor organs were deidentified by the organ procurement organization prior to shipment to the Alberta Diabetes Institute Islet-Core, and no identifying donor information was made available to the research team. A total of three nondiabetic (ND) donors were examined in this study. Full details of donor information, organ processing, and quality control information can be assessed with donor number (donors R224, R225, and R226 in this study) at https://www.isletcore.ca.

## Patch-clamp electrophysiology in human and mouse islets

Patch-clamp measurement of exocytic responses in mouse β-cells was performed as previously described (*Marcheva et al., 2020*). Dispersed human islets were cultured in low glucose (5.5 mM) DMEM media (supplemented with L-glutamine, 110 mg/l sodium pyruvate, 10% FBS, and 100 U/ml penicillin/streptomycin) in 35 mm culture dishes overnight. On the day of patch-clamp measurements, human or mouse islet cells were preincubated in extracellular solution at 1 mM glucose for 1 hr and capacitance was measured at 10 mM glucose with DMSO or 10 µM IVM as previously described (*Marcheva et al., 2020*). Mouse β-cells were identified by cell size and by half-maximal inactivation of $Na^+$ currents near –90 mV, and human β-cells were identified by immunostaining for positive insulin, following the experiment as described (*Fu et al., 2019*). Data analysis was performed using GraphPad Prism (v8.0c). Comparison of multiple groups was done by one- or two-way ANOVA, followed by Bonferroni or Tukey's post test. Data are expressed as means ± SEM, where $p < 0.05$ is considered significant.

## Single-cell RNA-seq analysis

Sequencing data from the study under SRA accession ERP017126 (*Segerstolpe et al., 2016*) were downloaded and converted to fastq files using the commands 'prefetch' followed by 'fastq-dump' through the sra-toolkit (v2.10.5). Each individual cell was aligned and transcript abundance quantified using RSEM with Hg38 (GRCh38.p12) as a reference. Raw single-cell expression count values were imported into RStudio for analysis using Seurat (*Hao et al., 2021*). Following low-quality cell removal, normalized expression values were used in uniform manifold approximation and projection (UMAP) dimensional reduction analyses to cluster distinct cell types. The R script, raw count tables, and parameters of these analyses are made publicly available under the Gene Expression Omnibus accession GSE186469.

## Western blotting

Beta-TC-6 cells lysates were isolated by treating cell pellets with RIPA buffer (Sigma-Aldrich) supplemented with 1× protease and 1× phosphatase inhibitors (Roche, Basel, Switzerland). Protein levels were quantified using Quick Start Bradford Protein Assay, and protein extracts were subject to SDS-PAGE gel electrophoresis and transferred to nitrocellulose membranes (Bio-Rad). Primary antibodies used were anti-P2Y1 (Santa Cruz, sc-377324) and anti-β-actin (Cell Signaling, Danvers, MA, CST 4970).

## Statistical analysis

Results were expressed as mean ± SEM unless otherwise noted. Information on sample size, genotype, and p values is provided within each figure and figure legend. Statistical significance of

capacitance, Fura2, and perifusion data was performed using a two-way ANOVA or mixed effects model (for datasets with missing values) with repeated measures followed by multiple comparison tests using a Bonferroni p-value adjustment via Prism (v9.2.0). Statistical analysis was performed by unpaired two-tailed Student's $t$-test unless otherwise indicated. $p<0.05$ was considered statistically significant. JTK_Cycle (v3) was used to determine rhythmicity in qPCR data using a period length of 24 hr and considering a Benjamini–Hochberg (BH)-adjusted $p$-value$<0.05$ as statistically rhythmic (*Hughes et al., 2010*).

## Acknowledgements

We thank all members of the Bass laboratory, Dr. Grant Barish, Dr. Lisa Beutler, and Dr. Richard Miller for helpful discussions and comments on the manuscript. We also thank Shun Kobayashi for technical assistance and Dr. Clara Bien Peek for the $Cry1^{-/-};Cry2^{-/-}$ mice. We thank the Human Organ Procurement and Exchange (HOPE) program and Trillium Gift of Life Network (TGLN) for their work in procuring human donor pancreas for research. Finally, we especially thank the organ donors and their families for their kind gift in support of diabetes research.

## Additional information

### Competing interests

Mark Perelis: Mark Perelis is affiliated with Ionis Pharmaceuticals, Inc The author has no financial interests to declare. The other authors declare that no competing interests exist.

### Funding

| Funder | Grant reference number | Author |
|---|---|---|
| National Institute of Diabetes and Digestive and Kidney Diseases | R01DK090625 | Joseph Bass |
| National Institute of Diabetes and Digestive and Kidney Diseases | R01DK127800 | Joseph Bass |
| National Institute of Diabetes and Digestive and Kidney Diseases | R01DK050203 | Joseph Bass |
| National Institute of Diabetes and Digestive and Kidney Diseases | R01DK113011 | Joseph Bass |
| National Institute on Aging | P01AG011412 | Joseph Bass |
| National Institute on Aging | R01AG065988 | Joseph Bass |
| Juvenile Diabetes Research Foundation United States of America | 17-2013-511 | Joseph Bass |
| Chicago Biomedical Consortium | S-007 | Joseph Bass |
| National Institute of Diabetes and Digestive and Kidney Diseases | DK007169 | Biliana Marcheva |
| National Institute of Diabetes and Digestive and Kidney Diseases | F30DK116481 | Benjamin J Weidemann |
| Manpei Suzuki Diabetes Foundation | | Akihiko Taguchi |
| Sino-Canadian Studentship | | Haopeng Lin |

| Funder | Grant reference number | Author |
|---|---|---|
| Canadian Institutes of Health Research | CIHR: 148451 | Patrick E Macdonald |

The funders had no role in study design, data collection and interpretation, or the decision to submit the work for publication.

## Author contributions

Biliana Marcheva, Conceptualization, Data curation, Formal analysis, Funding acquisition, Investigation, Methodology, Project administration, Supervision, Validation, Visualization, Writing – original draft, Writing – review and editing; Benjamin J Weidemann, Akihiko Taguchi, Mark Perelis, Conceptualization, Data curation, Formal analysis, Funding acquisition, Investigation, Methodology, Validation, Visualization, Writing – original draft, Writing – review and editing; Kathryn Moynihan Ramsey, Conceptualization, Data curation, Formal analysis, Investigation, Methodology, Supervision, Validation, Visualization, Writing – original draft, Writing – review and editing; Marsha V Newman, Conceptualization, Investigation, Methodology, Supervision, Visualization, Writing – original draft, Writing – review and editing; Yumiko Kobayashi, Chiaki Omura, Investigation, Methodology; Jocelyn E Manning Fox, Formal analysis, Investigation, Methodology; Haopeng Lin, Formal analysis, Funding acquisition, Investigation, Methodology, Writing – original draft, Writing – review and editing; Patrick E Macdonald, Formal analysis, Funding acquisition, Investigation, Methodology, Supervision, Writing – original draft, Writing – review and editing; Joseph Bass, Conceptualization, Funding acquisition, Investigation, Methodology, Project administration, Supervision, Validation, Visualization, Writing – original draft, Writing – review and editing

## Author ORCIDs

Biliana Marcheva (iD) http://orcid.org/0000-0001-8697-2625
Benjamin J Weidemann (iD) http://orcid.org/0000-0002-3747-2744
Kathryn Moynihan Ramsey (iD) http://orcid.org/0000-0002-0691-7835
Joseph Bass (iD) http://orcid.org/0000-0002-1602-8601

## Ethics

Human islet isolations and human islet cell biology experiments were performed at the Alberta Diabetes Institute IsletCore and approved by the University of Alberta Human Research Ethics Board (approval identifiers: Pro00013094, Pro00001754). Organ donation was coordinated by the appropriate regional organ procurement organization, including obtaining written next-of-kin consent for use of donor organs in this study. Donor organs were de-identified by the organ procurement organization prior to shipment to the Alberta Diabetes Institute IsletCore, and no identifying donor information was made available to the research team.

All animal care and use procedures were conducted in accordance with regulation of the Institutional Animal Care and Use Committee at Northwestern University under protocols IS00000466, IS00003253, IS00008732, IS0005838.

## Decision letter and Author response

Decision letter https://doi.org/10.7554/eLife.75132.sa1
Author response https://doi.org/10.7554/eLife.75132.sa2

# Additional files

## Supplementary files
• Supplementary file 1. High-throughput screen results.

• Supplementary file 2. Results of differential expression analysis in ivermectin (IVM)-treated WT and $P2ry1^{-/-}$ β cells.

• Transparent reporting form

## Data availability

Data in this study is publicly available in the GEO repository GSE186469. Source data files have been provided for the compounds used in the screen (Table S1), RNA-seq results (Table S2), and gel images (Source Data 1-2).

The following dataset was generated:

| Author(s) | Year | Dataset title | Dataset URL | Database and Identifier |
|---|---|---|---|---|
| Weidemann BJ | 2021 | High-throughput screen reveals puringeric receptor as a therapeutic target in circadian β-cell failure | https://www.ncbi.nlm.nih.gov/geo/query/acc.cgi?acc=GSE186469 | NCBI Gene Expression Omnibus, GSE186469 |

The following previously published datasets were used:

| Author(s) | Year | Dataset title | Dataset URL | Database and Identifier |
|---|---|---|---|---|
| Palasantza A, Sandberg R, Segerstolpe A | 2016 | Single-cell RNA-seq analysis of human pancreas from healthy individuals and type 2 diabetes patients | https://www.ebi.ac.uk/arrayexpress/experiments/E-MTAB-5061/ | ArrayExpress, ERP017126 |
| Perelis M, Marcheva B, Barish GD, Bass J | 2015 | Genome-wide Circadian Control of Transcription at Active Enhancers Regulates Insulin Secretion and Diabetes Risk | https://www.ncbi.nlm.nih.gov/geo/query/acc.cgi?acc=GSE69889 | NCBI Gene Expression Omnibus, GSE69889 |
| Perelis M | 2020 | A role for alternative splicing in circadian control of insulin secretion and glucose homeostasis | https://www.ncbi.nlm.nih.gov/geo/query/acc.cgi?acc=GSE146916 | NCBI Gene Expression Omnibus, GSE146916 |
| Palasantza A, Sandberg R, Segerstolpe A | 2016 | Single-cell RNA-seq analysis of human pancreas from healthy individuals and type 2 diabetes patients | https://www.ebi.ac.uk/arrayexpress/experiments/E-MTAB-5061/ | ArrayExpress, E-MTAB-5061 |

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
