## [Editor Report]

Circadian disruption is widespread in our modern 24/7 society, leading to an increased prevalence of common diseases including type 2 diabetes. The authors conducted an unbiased screen for small-molecule compounds that can restore the attenuated insulin secretion from pancreatic β-cells caused by a disrupted circadian clock. They identified ivermectin and its clock-controlled target, the P2Y1 receptor, which regulate glucose-stimulated ca^2+^ influx and insulin secretion in β-cells. This discovery represents an important advance in our understanding of regulatory mechanisms of insulin secretion by cell-autonomous clocks in mouse and human β-cells and is of fundamental clinical importance in context of novel therapeutic targets for diabetes management.

---

## [Decision Letter]

**Decision letter after peer review:**

Thank you for submitting your article "Pharmacologic rescue of circadian β-cell failure through P2Y1 purinergic receptor identified by small-molecule screen" for consideration by *eLife*. Your article has been reviewed by 2 peer reviewers, and the evaluation has been overseen by a Reviewing Editor and Didier Stainier as the Senior Editor. The following individual involved in review of your submission has agreed to reveal their identity: Charna Dibner (Reviewer #1).

Essential revisions:

1) Analyze the specificity of ivermectin for modulating insulin secretion regarding Bmal1-KO and β-cells, respectively. Test ivermectin on other endocrine cell types and other models for clock disruption.

2) Perform control experiments as requested by reviewer #2, major point 4, in particular those for experiments described in Figure 3 and Figure 4.

3) Discuss novelty compared to Burns et al., 2015.

*Reviewer #2 (Recommendations for the authors):*

Type 2 diabetes (T2D) is a prevalent metabolic disease that is phenotypically characterized by the loss of ß cell secretory function and cell mass. Its pathophysiology is complex and includes a combination of genetic and environmental risk factors. Studies in humans suggest that circadian misalignment (e.g. under shift work schedules) and disrupted sleep are associated with an increased incidence of T2D, as well as signs of ß cell failure and insulin insensitivity. In addition, mouse studies have confirmed that deletions of key components of the molecular clock disrupt normal glucose homeostasis, ß cell function, and insulin secretion. Global Bmal1 KO and pancreatic Bmal1 KO have been shown to negatively impact glucose tolerance, glucose stimulated insulin secretion, and pancreatic islet size. However, despite a relatively clear connection between circadian disruption and risk for development of T2D, molecular mechanisms are yet to be described in detail and hold potential for contributing to the development of novel therapeutic and preventative strategies.

The authors aim to tackle these questions by identifying pharmacological compounds that improve the phenotype of circadian ß cell failure, as well as by further describing the mode of action of their final target compound.

1) Do the authors believe that the pharmacological compound hit they identify in their screen is specific for Bmal1 KO induced ß cell failure or is the disruption of circadian rhythmicity more important than knock-out of this specific gene? Since they have the insulin-nanoluciferase reporter and the compound library they could perform a smaller sub-screen with selected candidates in a model of circadian ß cell failure that is different from Bmal1 KO.

2) My impression was that the authors are interested in finding pharmacological compounds that can rescue circadian ß cell failure. However, many of their results show that ivermectin also enhances insulin secretion in wild type cells/islets. So what is the conclusion of the authors? Did they just find "another drug for treatment of ß cell failure" or is it really specific for ß cell failure originating from circadian disruption (or perhaps Bmal1 KO)? If the former is the case, what would be the benefit of performing a compound screen in Bmal1 KO ß cells rather then in other established models of ß cell failure or wild type ß cells?

Specifically, I am referring to Figures Figure 3: Here the authors show that ivermectin treatment enhances glucose dependent insulin secretion, calcium influx, and membrane capacitance also in wild type islets. Why did they not test the effect of ivermectin on calcium influx and on insulin secretion in the perifusion system using their Bmal1 mutant islets?

Generally the effect of ivermectin in wild type cells/islets appears somewhat inconsistent. For example, it does not enhance membrane capacitance in wild type mouse islets but it does enhance it in human islets. In Figure 3D glucose + ivermectin treatment completely rescues insulin release in Bmal1 mutant islets (release is similar to wild type levels) but in Figure 4D it does not. Can the authors discuss this?

3) In their study from 2015, Burns et al., (Cell Metabolism, reference 76.) developed a insulin-gaussia luciferase reporter that to my understanding is identical to the insulin-nanoluciferase reporter except for exchange of the luciferase enzyme. Burns et al., also used their reporter system to screen for modifiers of insulin secretion in a high-throughput format. Could the authors please elaborate why they chose to replace gaussia by nanoluciferase and what is novel/beneficial about their approach?

4) Some important experimental controls are missing in some of the experiments (as follows). Also it is unclear why for some experiments the authors used 2mM "basal glucose" as control condition and sometimes they used 0mM glucose. Could they please explain?

Figure 3B – why are Bmal1 KO islets not included?

Figure 3C – again, why no Bmal1 KO islets?

Figure 4B – why is there no control with no glucose and just MRS?

Figure 4C – again, why is there no control with just MRS?

Figure S2D – 0mM glucose + Isr is missing

Figure S2E – again control with just Isr is missing

Figure S3B – P2ry1 transcript is rhythmic in wild type ß cells but is it's rhythmicity really lost in Bmal1 KO ß cells? (if the authors claim that P2ry1 is regulated by BMAL1 they should check this)

5) Could the authors elaborate on their strategy to identify potential ivermectin targets? It appears that ivermectin also augments insulin secretion in wild type cells, so is it really logical to check for transcripts that are already highly expressed in wild type? Wouldn't it be more intuitive to look for the top differentially expressed genes in wild type vs. Bmal1 KO ß cells? Also I would like to point out that the display of the differential gene expression data (Figure S3A) is difficult to understand for the reader. Rather than plotting the p-value the authors could show the log-fold change of expression (or similar) so that the reader can comprehend Bmal1 KO induced changes.

---

## [Author Response]

Essential revisions:1) Analyze the specificity of ivermectin for modulating insulin secretion regarding Bmal1-KO and β-cells, respectively. Test ivermectin on other endocrine cell types and other models for clock disruption.

We thank the Reviewers for these helpful suggestions. In our revised manuscript, we include new data from clock-disrupted cryptochrome (*Cry1^-/-^; Cry2^-/-^*) knockout mice. We show that islets isolated from these mice display impaired glucose-stimulated insulin secretion yet remain responsive to ivermectin (IVM) (new Figure 3D), similar to our findings in pancreas-specific *Bmal1^-/-^* mice (Figure 3B). Inclusion of this new data suggests that the restoration of insulin secretion in *Bmal1^-/-^* cells by IVM is due to rescue of a defect caused by the circadian clock network as opposed to a non-circadian effect of deletion of a specific transcription factor.

p7. We have added the following text: *“*Additionally, we observed a similar 2.9-fold increase in GSIS following administration of IVM to islets isolated from an independent mouse model of circadian disruption (Cry1^-/-^; Cry2^-/-^ mice) (Figure 3D), suggesting that IVM ameliorates secretory defects caused by disruption of the circadian clock network.*”*

We also appreciate the suggestion to assess the possible effect of IVM on other endocrine cell types within the islets. Since our screen was performed in an exclusively β-cell culture, it is unlikely that IVM effects on other endocrine cell types, particularly glucagon-secreting α cells, explain the insulinotropic effects of IVM in whole islets studies. While we cannot rule out effects of IVM on other cell types, we now provide data from our new single-cell RNA-sequencing analysis (SRA accession ERP017126) in human islets that shows that *P2RY1* expression is enriched in insulin-producing β cells but not within glucagon-expressing α cells (new Figure 4—figure supplement 1C) (1). Therefore, we do not believe that IVM action through P2Y1 within α cells contributes to the enhanced insulin secretion observed in these islets.

p8. We have added the following text*:* “Finally, analysis of RNA-sequencing data from human islets (SRA accession ERP017126) indicates that P2RY1 expression is enriched within β cells among hormone-secreting cell types, with little to no detectable expression in the glucagon-secreting α cells (Figure 4—figure supplement 1C).”

2) Perform control experiments as requested by reviewer #2, major point 4, in particular those for experiments described in Figure 3 and Figure 4.

In our revised manuscript, we now include the controls requested by Reviewer 2 in Major Comment 4.

(i) A prior critique was the lack of a dynamic assessment of insulin secretion in IVM-treated *Bmal1^-/-^* islets. In response, we have now performed dynamic perifusion experiments, where we measure insulin every 2 min over the course of 30 min following stimulation with either 20 mM glucose or 20 mM glucose plus 10 µM IVM. While *Bmal1^-/-^* islets exhibit reduced glucose-stimulated insulin release, IVM enhances *Bmal1^-/-^* islet insulin secretion to the same level observed in islets from control mice not exposed to drug during both the initial burst of insulin secretion (first 12 min) and the sustained release (12 – 30 min). These results provide new dynamic evidence for IVM-augmentation of glucose-stimulated insulin secretion in the setting of molecular clock disruption (new Figure 3C).

p7. We have added the following text: “Furthermore, perifusion experiments in islets from Bmal1 mutant mice revealed that IVM significantly increased insulin release during both the initial burst of secretion (first 12 minutes post-stimulation) and during the sustained release (12-30 min) in both WT and Bmal1 mutant islets (Figure 3C).”

(ii) We present new control data demonstrating that the P2Y1 inhibitor MRS2179 (MRS) has no effect on insulin secretion or calcium influx in the absence of glucose (new Figure 4B-C).

(iii) We also show new control data demonstrating that the calcium channel inhibitor isradipine (Isr) decreases insulin secretion and calcium influx regardless of the presence of glucose or IVM, suggesting IVM acts upstream of calcium channel signaling (new Figure 3—figure supplement 1C-D).

(iv) Finally, we also present new requested data demonstrating that *P2ry1* levels remain reduced across the 24-hr day, and that rhythmic expression is lost in synchronized *Bmal1^-/-^* pseudoislets (new Figure 4—figure supplement 1B). This new data, in combination with our previous genomics evidence for enrichment of BMAL1 binding sites within the *P2ry1* enhancer region (Figure 4A, Figure 4—figure supplement 1A), support a direct role for BMAL1 in regulation of *P2ry1*.

p8. We have added the following text: “We also found decreased levels and loss in rhythmicity of P2ry1 in synchronized Bmal1^-/-^ pseudoislets (Figure 4—figure supplement 1B). BMAL1 chromatin immunoprecipitation-sequencing in Β-TC-6 cells also revealed enrichment of BMAL1 chromatin binding within enhancer regions 266 – 41 kb upstream of the P2ry1 gene transcription start site (GSE69889) (Figure 4A, Figure 4—figure supplement 1A).”

3) Discuss novelty compared to Burns et al., 2015.

We thank the reviewers for the opportunity to expand upon the novelty of our screen in reference to previous publications, including that of Burns et al., 2015 (2). We designed our screen to identify small molecules and pharmacologic targets that can augment glucose-coupled insulin secretion in the context of impaired β cell function following abrogation of the molecular clock. In contrast, Burns et al., sought to identify compounds that activate or repress glucose-coupled insulin secretion in two strains of wild-type β cells in the absence of disease. Since molecular pathways that drive insulin secretion are perturbed during disease (3-5), we sought to identify compounds and pathways that effectively augment insulin secretion despite alterations in molecular pathways present in the diseased state. While this does not exclude compounds that effectively enhance insulin secretion in the “healthy” cell, we reasoned that our phenotype-driven approach would increase the sensitivity to identify targets with high predictive validity. Therefore, our concept was to leverage circadian mutant β cells as a rare (monogenic) form of diabetes, with the goal of uncovering insulinotropic mechanisms downstream of the molecular clock and at the same time finding targets that may have broad applicability as anti-hyperglycemic agents. We believe that our effort was fruitful insofar as our studies have shown insulinotropic effects of IVM in three independent models of β-cell failure, including in *Bmal1* KO, *Cry* double KO, and *Akita* diabetic mice (Figure 3B-D, Figure 3—figure supplement 1F-G).

To directly compare the results of our study with those found in Burns et al., we tabulated Z-scores of similarly annotated compounds side-by-side between our screen and those performed in INS1E and MIN6 cells in Burns et al., 2015 (Author response image 1). Of note, two lead compounds (ivermectin and tyrothricin) in Burns et al., had a positive Z-score in both INS1E and MIN6 cells, consistent with findings from our study (Figure 2B-D). In contrast, suloctidil was found to be insulinotropic in our screen but insulin-repressive in the Burns study. Further, while the compounds bufexamac, doramectin, moxidectin, and lobendazole were efficacious in promoting insulin secretion in wild-type MIN6 cells, these were found to be either ineffective (Z-score < 1) or inhibitory for insulin secretion in BMAL1-ablated cells, which suggests these compounds may be ineffective in treating diabetes with a circadian component. Finally, potent inhibitors of insulin secretion in both screens included the calcium channel blockers isradipine and nitrendipine, consistent with a role for ca^2+^ influx in β-cell physiology (6, 7). Together, these results highlight the importance of studying compounds in a context where the assay model (circadian disruption) reflects the relevant disease (type 2 diabetes).

**Author response image 1. sa2fig1:** Comparison of our circadian small molecule screen to Burns et al. , 2015 screen. The hits from each screen were compared by analyzing Z-scores, which reflect the compounds’ insulin-secretory effects on *Bmal1^-/-^* Β-TC-6 cells (our screen) versus the cell-types used by Burns et al., wild-type MIN6 (left) and INS1E cells (right), in the presence of glucose.

Reviewer #2 (Recommendations for the authors):Type 2 diabetes (T2D) is a prevalent metabolic disease that is phenotypically characterized by the loss of ß cell secretory function and cell mass. Its pathophysiology is complex and includes a combination of genetic and environmental risk factors. Studies in humans suggest that circadian misalignment (e.g. under shift work schedules) and disrupted sleep are associated with an increased incidence of T2D, as well as signs of ß cell failure and insulin insensitivity. In addition, mouse studies have confirmed that deletions of key components of the molecular clock disrupt normal glucose homeostasis, ß cell function, and insulin secretion. Global Bmal1 KO and pancreatic Bmal1 KO have been shown to negatively impact glucose tolerance, glucose stimulated insulin secretion, and pancreatic islet size. However, despite a relatively clear connection between circadian disruption and risk for development of T2D, molecular mechanisms are yet to be described in detail and hold potential for contributing to the development of novel therapeutic and preventative strategies.The authors aim to tackle these questions by identifying pharmacological compounds that improve the phenotype of circadian ß cell failure, as well as by further describing the mode of action of their final target compound.1) Do the authors believe that the pharmacological compound hit they identify in their screen is specific for Bmal1 KO induced ß cell failure or is the disruption of circadian rhythmicity more important than knock-out of this specific gene? Since they have the insulin-nanoluciferase reporter and the compound library they could perform a smaller sub-screen with selected candidates in a model of circadian ß cell failure that is different from Bmal1 KO.

The Reviewer raises the important question about whether IVM-enhanced GSIS in the *Bmal1^-/-^* β cells is specific to loss of the BMAL1 transcription factor itself, or whether IVM rescues GSIS in other models of clock network disruption. As described above in the response to Essential Revisions Point #1, we provide new data demonstrating that IVM also enhances GSIS in an independent model of circadian disruption (*Cry1^-/-^; Cry2^-/-^*) (new Figure 3D in response to Essential Revisions Point #1), in addition to *Bmal1^-/-^* cells. Inclusion of this new data suggests that the restoration of insulin secretion in the *Bmal1^-/-^* cells by IVM is due to rescue of a defect caused by the circadian clock network itself as opposed a non-circadian effect of deletion of a specific transcription factor. We have added this new data to the manuscript as described above in the response to Essential Revisions Point #1.

2) My impression was that the authors are interested in finding pharmacological compounds that can rescue circadian ß cell failure. However, many of their results show that ivermectin also enhances insulin secretion in wild type cells/islets. So what is the conclusion of the authors? Did they just find "another drug for treatment of ß cell failure" or is it really specific for ß cell failure originating from circadian disruption (or perhaps Bmal1 KO)? If the former is the case, what would be the benefit of performing a compound screen in Bmal1 KO ß cells rather then in other established models of ß cell failure or wild type ß cells?

The Reviewer raises an interesting point regarding our finding that IVM enhances insulin secretion in the setting of both circadian disruption and in wild-type β cells. In our revised manuscript, we clarify that we specifically designed our screen to identify small molecules and pharmacologic targets that can augment glucose-coupled insulin secretion in the context of impaired β cell function following abrogation of the molecular clock. We chose this phenotype-driven approach to identify compounds and pathways that effectively augment insulin secretion despite alterations in molecular pathways present in the diseased state, with the goal of uncovering insulinotropic mechanisms downstream of the molecular clock and at the same time finding targets that may have broad applicability as anti-hyperglycemic agents.

We note that this does not exclude identification of compounds that effectively enhance insulin secretion in the “healthy” cell (as the Reviewer noted). The finding that IVM enhances glucose-stimulated insulin secretion in wild-type islets in addition to *Bmal1* KOs does not diminish the novel observation that IVM is able to augment insulin secretion during clock disruption. We identified pharmacologic activation of the P2Y1 receptor as a potential therapeutic avenue by using a genetic-sensitized screen, combined with genomics analyses in the *Bmal1* KO islets. Since the phenotype of the circadian mutant β cells used in this screen overlaps with features of β-cell failure found in T2D, we expect the hits from this screen to be more predictive of therapeutic effect in diabetes relative to those identified in healthy β cells. Supporting this hypothesis, we found IVM to be efficacious in enhancing glucose-responsive insulin secretion in multiple models of insulin secretion defects, including circadian-disrupted *Cry* double KOs and *Akita* islets. Future studies on the transcriptional networks shared between these models and how BMAL1::CLOCK/NPAS2 regulate physiological activity of purinergic receptors are warranted.

An additional perspective as to the utility of the genetically-sensitized circadian mutant screen is that many insulinotropic factors in WT cells, such as those directly inducing depolarization via closure of the potassium sulfonylurea channel, would not stimulate insulin secretion in the setting of circadian mutant cells (12)—in essence the approach we’ve taken of using mutant cells as a first level of screening provides a “filter” to enhance detection of “hit” compounds that might work in mutant cells and likewise reveal mechanistic insight into the circadian basis for β-cell function. We speculate that phenotype-driven screens similar to our present study can in the future expand using cell-based assays as a starting point to identify insulinotropic compounds and mechanisms involved in rare genetic (i.e. MODY mutations) and even more common environmental (i.e. lipoglucotoxicity models) conditions.

Specifically, I am referring to Figures Figure 3: Here the authors show that ivermectin treatment enhances glucose dependent insulin secretion, calcium influx, and membrane capacitance also in wild type islets. Why did they not test the effect of ivermectin on calcium influx and on insulin secretion in the perifusion system using their Bmal1 mutant islets?

We now provide new data demonstrating a decrease in glucose-stimulated insulin release during dynamic perifusion experiments in *Bmal1^-/-^* islets that is restored in the presence of IVM, supporting a role for IVM-augmentation of glucose-stimulated insulin secretion in a model of clock disruption (see detailed response to Essential Revisions Point #1, new Figure 3C).

Generally the effect of ivermectin in wild type cells/islets appears somewhat inconsistent. For example, it does not enhance membrane capacitance in wild type mouse islets but it does enhance it in human islets. In Figure 3D glucose + ivermectin treatment completely rescues insulin release in Bmal1 mutant islets (release is similar to wild type levels) but in Figure 4D it does not. Can the authors discuss this?

Our insulin secretion and calcium influx experiments consistently show that IVM increases secretion and calcium flux in WT mouse cells and islets, as well as in islets from control mice in our *Bmal1* and *Cry* mutant experiments (Figures 2D, 3A-D 4B-D, Figure 3—figure supplement 1A-C). IVM treatment also enhances membrane capacitance in human islets (Figure 3F) though as the Reviewer noted, IVM did not result in a significant increase in membrane capacitance in control mouse islets (Figure 3E), which could be a result of either a small sample size (n=4 experiments with IVM in controls) or alternatively that capacitance under these conditions is already maximally stimulated in control islets, thereby preventing a further increase in capacitance following IVM stimulation.

We further note that our experiments also consistently show that IVM increases insulin release and calcium flux in circadian mutant cells and islets (Figures 3B-E, 4D), suggesting IVM can consistently rescue impaired secretory capacity in circadian mutants to physiological/wild-type levels, though the Reviewer notes there are differences in the degree of stimulation in the islets from *Bmal1* mutant mice in Figures 3B,D vs the pseudoislets from *Bmal1* mutant Β-TC-6 cells in Figure 4D, for example. Given our evidence that IVM augments P2Y1 signaling, it is possible that islet cell-to-cell paracrine nucleosides (13) may be required for full IVM effect. Therefore, the differences in degree of stimulation in islets versus pseudoislets may represent a difference in β-cell versus mixed-islet-cell response to IVM. Future studies will seek to understand IVM effects on islet cell crosstalk and other islet hormones.

3) In their study from 2015, Burns et al., (Cell Metabolism, reference 76.) developed a insulin-gaussia luciferase reporter that to my understanding is identical to the insulin-nanoluciferase reporter except for exchange of the luciferase enzyme. Burns et al., also used their reporter system to screen for modifiers of insulin secretion in a high-throughput format. Could the authors please elaborate why they chose to replace aussian by nanoluciferase and what is novel/beneficial about their approach?

We thank the Reviewer for the opportunity to clarify the novelty of our approach compared to a previously published small molecule screens for insulinotropic compounds (Burns et al., 2015) (2). We first wish to clarify that we used a Proinsulin-NanoLuc lentiviral vector (Proinsulin-NanoLuc in pLX304) that was a gift from David Altshuler, senior author of the Burns et al., paper (Addgene plasmid #62057; http://n2t.net/addgene:62057). The plasmid we received contains a Nano-luciferase insert as opposed to the Gaussia luciferase insert reported in Burns et al., 2015. For this reason, we independently validated our screen against an insulin immunoassay (Figure 1C). We note that Nanoluciferase is a slightly smaller construct (19 kDa) than the Gaussian luciferase (20 kDa), but both emit light in response to the compound substrate (i.e. Nano-Glo Substrate, Promega) in an ATP-independent manner.

Regarding novelty of our approach, as described above in the Essential Revisions Point #3, we designed our screen to identify small molecules and pharmacologic targets that can augment glucose-coupled insulin secretion in the context of impaired β cell function following abrogation of the molecular clock. Burns et al., on the other hand, sought to identify compounds that either activate or repress glucose-coupled insulin secretion in two strains of wild-type β cells in the absence of disease. Thus, we sought to identify compounds and pathways that effectively augment insulin secretion despite alterations in molecular pathways present in the diseased state. While this does not exclude compounds that effectively enhance insulin secretion in the “healthy” cell, we reasoned that our phenotype-driven approach would increase the sensitivity to identify targets with high predictive validity. Therefore, our concept was to leverage circadian mutant β cells as a rare (monogenic) form of diabetes, with the goal of uncovering insulinotropic mechanisms downstream of the molecular clock and at the same time finding targets that may have broad applicability as anti-hyperglycemic agents. We believe that our effort was fruitful insofar, as our studies showed insulinotropic effects of IVM in three independent models of β-cell failure, including *Bmal1* KO and *Cry* double KO mice as well as in the setting of the *Akita* diabetic mouse (Figure 3B-D, Figure 3—figure supplement 1F-G).

Finally, to compare the results of our study with those found in Burns et al., we tabulated Z-scores of similarly annotated compounds side-by-side between our screen and the screens performed using INS1E and MIN6 cells in Burns et al., 2015 (see Author response image 1). Of note, two lead compounds (ivermectin and tyrothricin) in Burns et al., had a positive Z-score in both INS1E and MIN6 cells, consistent with findings from our study. In contrast, suloctidil was found to be insulinotropic in our screen but insulin-repressive in the Burns study. Further, while the compounds bufexamac, doramectin, moxidectin, and lobendazole were efficacious in promoting insulin secretion in wild-type MIN6 cells, these were found to be either ineffective (Z-Score < 1) or inhibitory for insulin secretion in BMAL1-ablated cells, which suggests these compounds may be ineffective in treating diabetes with a circadian component. Finally, potent inhibitors of insulin secretion in both screens included the calcium channel blockers isradipine and nitrendipine, consistent with a role for ca^2+^ influx in β-cell physiology (6, 7). Together, these results highlight the importance of studying compounds in a context where the assay model (circadian disruption) reflects the relevant the disease (type 2 diabetes).

4) Some important experimental controls are missing in some of the experiments (as follows).

We now include the below-mentioned controls in our revised manuscript:

Figure 3B – why are Bmal1 KO islets not included?Figure 3C – again, why no Bmal1 KO islets?

We now show new data demonstrating a decrease in glucose-stimulated insulin release during dynamic perifusion experiments in *Bmal1* KO pseudoislets that is restored in the presence of IVM, supporting a role for IVM-augmentation of glucose-stimulated insulin secretion in a model of clock disruption (see detailed response to Essential Revisions Point #2 and new Figure 3C). We found that BMAL1-deficient β cells fail to appropriately respond to glucose in the context of the calcium influx assay performed in the original Figure 3B, which utilizes a nutrient deplete media; thus continued work to optimize and identify different key factors influencing wild-type versus circadian mutant performance in the context of calcium influx will be the subject of future studies.

Figure 4B – why is there no control with no glucose and just MRS?Figure 4C – again, why is there no control with just MRS?

We now present new data demonstrating the P2Y1 inhibitor MRS has no effect on insulin secretion or calcium influx in the absence of glucose (see detailed response to Essential Revisions Point #2 and new Figures 4B-C)

Figure S2D – 0mM glucose + Isr is missingFigure S2E – again control with just Isr is missing

We now include the requested 0 mM glucose + Isr control and reveal decreases in insulin secretion and calcium influx following calcium channel inhibition with Isr in the absence of glucose (see detailed response to Essential Revisions Point #2 and new Figure 3—figure supplement 1C-D)

Figure S3B – P2ry1 transcript is rhythmic in wild type ß cells but is it's rhythmicity really lost in Bmal1 KO ß cells? (if the authors claim that P2ry1 is regulated by BMAL1 they should check this)

We now present new data demonstrating that *P2ry1* levels remain reduced and arhythmic across the 24-hr day in synchronized *Bmal1^-/-^* pseudoislets (see detailed response to Essential Revisions Point #2 and new Figure 4—figure supplement 1B), supporting a direct role for BMAL1 in regulation of *P2ry1*.

Also it is unclear why for some experiments the authors used 2mM "basal glucose" as control condition and sometimes they used 0mM glucose. Could they please explain?

We have clarified in the methods that for the GSIS assays performed in primary islets, we used 2 mM glucose as the basal glucose concentration, which is standard in the field (14). We further clarified that 0 mM glucose was used as the basal glucose concentration for all GSIS assays performed in either β cell lines or pseudoislets in order to minimize the typically higher levels of unstimulated glucose-stimulated insulin secretion that occurs in β cell lines compared to islets (5).

5) Could the authors elaborate on their strategy to identify potential ivermectin targets? It appears that ivermectin also augments insulin secretion in wild type cells, so is it really logical to check for transcripts that are already highly expressed in wild type? Wouldn't it be more intuitive to look for the top differentially expressed genes in wild type vs. Bmal1 KO ß cells? Also I would like to point out that the display of the differential gene expression data (Figure S3A) is difficult to understand for the reader. Rather than plotting the p-value the authors could show the log-fold change of expression (or similar) so that the reader can comprehend Bmal1 KO induced changes.

Strategy to identify potential IVM targets*:* To identify potential IVM targets, we first performed a literature review to identify previously-published IVM targets to use as starting points, which included ion channels, G-protein coupled receptors, ionotropic receptors (such as purinergic, GABAergic, and glycine receptors), as well as farnesoid X nuclear receptors (15, 16). Our observations that IVM rapidly increases calcium flux and insulin release in a glucose-dependent manner argues against a role for a transcriptional regulator (FXR) or a direct ionophore effect (GABA) and is more consistent with a role for β-cell G-protein coupled receptor activation, which requires glucose for appropriate signaling (17, 18). Recently, P2Y1 was implicated in nutrient- and ATP/ADP-dependent regulation of insulin release through an adipocyte-islet axis, further suggesting P2Y1 may play a role in physiologic regulation of islet hormone release (13). Additional studies will be required to determine whether IVM affects paracrine ATP/ADP release to affect P2Y1 or whether IVM directly binds purinergic receptors in the β cell.

Analysis of differentially-expressed genes in BKOs to provide insight to potential candidates: Next, in agreement with the Reviewer’s suggestion, we considered top differentially-expressed genes in the WT vs *Bmal1^-/-^* β cells from our RNA-seq analyses. To clarify these findings in the text, we re-organized our Results section and now begin with data showing that *P2ry1* is one of the most differentially-expressed genes in the *Bmal1^-/-^* β cells, with a 3.1-fold reduction (Adj. P=10^-55^) in expression (Figure 4A). We have also updated Figure S3A as requested to include the log-fold change of *P2ry1* in addition to the p-value (Figure 4—figure supplement 1A, Author response image 2). Further support for BMAL1 control of *P2ry1* is enriched BMAL1 chromatin binding within enhancer regions 266-41 kb upstream of the *P2ry1* gene transcription start site by ChIP-sequencing (Figure 4A, Figure 4—figure supplement 1A), rhythmic expression of *P2ry1* in WT Β-TC-6 pseudoislets (Figure 4—figure supplement 1B), and new data showing loss of rhythmic *P2ry1* expression in *Bmal1^-/-^* pseudoislets (new Figure 4—figure supplement 1B).

**Author response image 2. sa2fig2:** Robust differential expression of *P2ry1* following BMAL1 ablation. The *P2ry1* transcript is one of the most abundant receptors in the broad class of putative IVM targets and was identified following BMAL1 ablation as one of the most consistently and highly repressed transcripts (Adj. P < 10^-55^).

Thus, since it is known that IVM augments purinergic signaling (19, 20), that the purinergic receptor P2Y1 is significantly reduced in *Bmal1^-/-^* β cells, and that BMAL1 specifically controls *P2ry1* amongst the purinergic receptor family in the β cell (Figure 4A, Figure 4—figure supplement 1A-B), we sought to test the functional role of the P2Y1 receptor in the insulinotropic action of IVM. While we cannot rule out potential action of IVM on other targets, evidence from our pharmacologic and genetic *P2ry1* ablation models suggest P2Y1 is a critical component of the IVM insulinotropic effect.

References

1. Segerstolpe A, Palasantza A, Eliasson P, Andersson EM, Andreasson AC, Sun X, Picelli S, Sabirsh A, Clausen M, Bjursell MK, Smith DM, Kasper M, Ammala C, Sandberg R. Single-Cell Transcriptome Profiling of Human Pancreatic Islets in Health and Type 2 Diabetes. Cell Metab. 2016;24(4):593-607. Epub 2016/09/27. doi: 10.1016/j.cmet.2016.08.020. PubMed PMID: 27667667; PMCID: PMC5069352.

2. Burns SM, Vetere A, Walpita D, Dancik V, Khodier C, Perez J, Clemons PA, Wagner BK, Altshuler D. High-throughput luminescent reporter of insulin secretion for discovering regulators of pancreatic Β-cell function. Cell Metab. 2015;21(1):126-37. Epub 2015/01/08. doi: 10.1016/j.cmet.2014.12.010. PubMed PMID: 25565210.

3. Perelis M, Marcheva B, Ramsey KM, Schipma MJ, Hutchison AL, Taguchi A, Peek CB, Hong H, Huang W, Omura C, Allred AL, Bradfield CA, Dinner AR, Barish GD, Bass J. Pancreatic β cell enhancers regulate rhythmic transcription of genes controlling insulin secretion. Science. 2015;350(6261):aac4250. doi: 10.1126/science.aac4250. PubMed PMID: 26542580; PMCID: 4669216.

4. Avrahami D, Wang YJ, Schug J, Feleke E, Gao L, Liu C, Consortium H, Naji A, Glaser B, Kaestner KH. Single-cell transcriptomics of human islet ontogeny defines the molecular basis of β-cell dedifferentiation in T2D. Mol Metab. 2020;42:101057. Epub 2020/08/03. doi: 10.1016/j.molmet.2020.101057. PubMed PMID: 32739450; PMCID: PMC7471622.

5. Marcheva B, Perelis M, Weidemann BJ, Taguchi A, Lin H, Omura C, Kobayashi Y, Newman MV, Wyatt EJ, McNally EM, Fox JEM, Hong H, Shankar A, Wheeler EC, Ramsey KM, MacDonald PE, Yeo GW, Bass J. A role for alternative splicing in circadian control of exocytosis and glucose homeostasis. Genes Dev. 2020;34(15-16):1089-105. Epub 2020/07/04. doi: 10.1101/gad.338178.120. PubMed PMID: 32616519; PMCID: PMC7397853.

6. Schulla V, Renstrom E, Feil R, Feil S, Franklin I, Gjinovci A, Jing XJ, Laux D, Lundquist I, Magnuson MA, Obermuller S, Olofsson CS, Salehi A, Wendt A, Klugbauer N, Wollheim CB, Rorsman P, Hofmann F. Impaired insulin secretion and glucose tolerance in β cell-selective Ca(v)1.2 ca^2+^ channel null mice. EMBO J. 2003;22(15):3844-54. Epub 2003/07/26. doi: 10.1093/emboj/cdg389. PubMed PMID: 12881419; PMCID: PMC169062.

7. Vasseur M, Debuyser A, Joffre M. Sensitivity of pancreatic β cell to calcium channel blockers. An electrophysiologic study of verapamil and nifedipine. Fundam Clin Pharmacol. 1987;1(2):95-113. Epub 1987/01/01. doi: 10.1111/j.1472-8206.1987.tb00549.x. PubMed PMID: 3315915.

8. Bult CJ, Blake JA, Smith CL, Kadin JA, Richardson JE, Mouse Genome Database G. Mouse Genome Database (MGD) 2019. Nucleic Acids Res. 2019;47(D1):D801-D6. Epub 2018/11/09. doi: 10.1093/nar/gky1056. PubMed PMID: 30407599; PMCID: PMC6323923.

9. Gorgani NN, Smith BA, Kono DH, Theofilopoulos AN. Histidine-rich glycoprotein binds to DNA and Fc γ RI and potentiates the ingestion of apoptotic cells by macrophages. J Immunol. 2002;169(9):4745-51. Epub 2002/10/23. doi: 10.4049/jimmunol.169.9.4745. PubMed PMID: 12391183.

10. Nauck M, Stockmann F, Ebert R, Creutzfeldt W. Reduced incretin effect in type 2 (non-insulin-dependent) diabetes. Diabetologia. 1986;29(1):46-52. Epub 1986/01/01. doi: 10.1007/BF02427280. PubMed PMID: 3514343.

11. Drucker DJ, Nauck MA. The incretin system: glucagon-like peptide-1 receptor agonists and dipeptidyl peptidase-4 inhibitors in type 2 diabetes. Lancet. 2006;368(9548):1696-705. Epub 2006/11/14. doi: 10.1016/S0140-6736(06)69705-5. PubMed PMID: 17098089.

12. Marcheva B, Ramsey KM, Buhr ED, Kobayashi Y, Su H, Ko CH, Ivanova G, Omura C, Mo S, Vitaterna MH, Lopez JP, Philipson LH, Bradfield CA, Crosby SD, Jebailey L, Wang X, Takahashi JS, Bass J. Disruption of the clock components CLOCK and BMAL1 leads to hypoinsulinaemia and diabetes. Nature. 2010;466(7306):571-2. Epub 2010/06/22. doi: nature09253 [pii]10.1038/nature09253. PubMed PMID: 20562852.

13. Prentice KJ, Saksi J, Robertson LT, Lee GY, Inouye KE, Eguchi K, Lee A, Cakici O, Otterbeck E, Cedillo P, Achenbach P, Ziegler AG, Calay ES, Engin F, Hotamisligil GS. A hormone complex of FABP4 and nucleoside kinases regulates islet function. Nature. 2021;600(7890):720-6. Epub 2021/12/10. doi: 10.1038/s41586-021-04137-3. PubMed PMID: 34880500.

14. Carter JD, Dula SB, Corbin KL, Wu R, Nunemaker CS. A practical guide to rodent islet isolation and assessment. Biol Proced Online. 2009;11:3-31. Epub 2009/12/04. doi: 10.1007/s12575-009-9021-0. PubMed PMID: 19957062; PMCID: PMC3056052.

15. Chen IS, Kubo Y. Ivermectin and its target molecules: shared and unique modulation mechanisms of ion channels and receptors by ivermectin. J Physiol. 2018;596(10):1833-45. Epub 2017/10/25. doi: 10.1113/JP275236. PubMed PMID: 29063617; PMCID: PMC5978302.

16. Dawson GR, Wafford KA, Smith A, Marshall GR, Bayley PJ, Schaeffer JM, Meinke PT, McKernan RM. Anticonvulsant and adverse effects of avermectin analogs in mice are mediated through the γ-aminobutyric acid(A) receptor. J Pharmacol Exp Ther. 2000;295(3):1051-60. Epub 2000/11/18. PubMed PMID: 11082440.

17. Gilon P, Henquin JC. Mechanisms and physiological significance of the cholinergic control of pancreatic β-cell function. Endocr Rev. 2001;22(5):565-604. Epub 2001/10/06. doi: 10.1210/edrv.22.5.0440. PubMed PMID: 11588141.

18. Leon C, Freund M, Latchoumanin O, Farret A, Petit P, Cazenave JP, Gachet C. The P2Y(1) receptor is involved in the maintenance of glucose homeostasis and in insulin secretion in mice. Purinergic Signal. 2005;1(2):145-51. Epub 2008/04/12. doi: 10.1007/s11302-005-6209-x. PubMed PMID: 18404499; PMCID: PMC2096536.

19. Bowler JW, Bailey RJ, North RA, Surprenant A. P2X4, P2Y1 and P2Y2 receptors on rat alveolar macrophages. Br J Pharmacol. 2003;140(3):567-75. Epub 2003/09/13. doi: 10.1038/sj.bjp.0705459. PubMed PMID: 12970084; PMCID: PMC1574050.

20. Hansen MR, Krabbe S, Novak I. Purinergic receptors and calcium signalling in human pancreatic duct cell lines. Cell Physiol Biochem. 2008;22(1-4):157-68. Epub 2008/09/05. doi: 10.1159/000149793. PubMed PMID: 18769042.

21. von der Kammer H, Demiralay C, Andresen B, Albrecht C, Mayhaus M, Nitsch RM. Regulation of gene expression by muscarinic acetylcholine receptors. Biochem Soc Symp. 2001(67):131-40. Epub 2001/07/13. doi: 10.1042/bss0670131. PubMed PMID: 11447829.

22. Mali P, Yang L, Esvelt KM, Aach J, Guell M, DiCarlo JE, Norville JE, Church GM. RNA-guided human genome engineering via Cas9. Science. 2013;339(6121):823-6. Epub 2013/01/05. doi: 10.1126/science.1232033. PubMed PMID: 23287722; PMCID: PMC3712628.

23. Ran FA, Hsu PD, Wright J, Agarwala V, Scott DA, Zhang F. Genome engineering using the CRISPR-Cas9 system. Nat Protoc. 2013;8(11):2281-308. Epub 2013/10/26. doi: 10.1038/nprot.2013.143. PubMed PMID: 24157548; PMCID: PMC3969860.

24. Hsu PD, Lander ES, Zhang F. Development and applications of CRISPR-Cas9 for genome engineering. Cell. 2014;157(6):1262-78. Epub 2014/06/07. doi: 10.1016/j.cell.2014.05.010. PubMed PMID: 24906146; PMCID: PMC4343198.